# Human ORC/MCM density is low in active genes and correlates with replication time but does not delimit initiation zones

Nina Kirstein[1†], Alexander Buschle[2], Xia Wu[3‡], Stefan Krebs[4], Helmut Blum[4], Elisabeth Kremmer[5], Ina M Vorberg[6,7], Wolfgang Hammerschmidt[2], Laurent Lacroix[3], Olivier Hyrien[3]*, Benjamin Audit[8]*, Aloys Schepers[1§]*

[1]Research Unit Gene Vectors, Helmholtz Zentrum München (GmbH), German Research Center for Environmental Health, Munich, Germany; [2]Research Unit Gene Vectors, Helmholtz Zentrum München (GmbH), German Research Center for Environmental Health and German Center for Infection Research (DZIF), Munich, Germany; [3]Institut de Biologie de l'ENS (IBENS), Département de Biologie, Ecole Normale Supérieure, CNRS, Inserm, PSL Research University, Paris, France; [4]Laboratory for Functional Genome Analysis (LAFUGA), Gene Center of the Ludwig-Maximilians Universität (LMU) München, Munich, Germany; [5]Institute for Molecular Immunology, Monoclonal Antibody Core Facility, Helmholtz Zentrum München, German Research Center for Environmental Health (GmbH), Neuherberg, Germany; [6]German Center for Neurodegenerative Diseases (DZNE e.V.), Bonn, Germany; [7]Rheinische Friedrich-Wilhelms-Universität Bonn, Bonn, Germany; [8]Univ Lyon, ENS de Lyon, Univ. Claude Bernard, CNRS, Laboratoire de Physique, 69342 Lyon, France

*For correspondence:
hyrien@biologie.ens.fr (OH);
benjamin.audit@ens-lyon.fr (BA);
schepers@helmholtz-muenchen.
de (AS)

Present address: † University of Miami, Miller School of Medicine, Sylvester Comprehensive Cancer Center, Department of Human Genetics, Miami, United States; ‡ Zhongshan School of Medicine, Sun Yat-sen University, Guangzhou, China; § Institute for Diabetes and Obesity, Monoclonal Antibody Core Facility, Helmholtz Zentrum München (GmbH), German Research Center for Environmental Health, Neuherberg, Germany

Competing interests: The authors declare that no competing interests exist.

**Abstract** Eukaryotic DNA replication initiates during S phase from origins that have been licensed in the preceding G1 phase. Here, we compare ChIP-seq profiles of the licensing factors Orc2, Orc3, Mcm3, and Mcm7 with gene expression, replication timing, and fork directionality profiles obtained by RNA-seq, Repli-seq, and OK-seq. Both, the origin recognition complex (ORC) and the minichromosome maintenance complex (MCM) are significantly and homogeneously depleted from transcribed genes, enriched at gene promoters, and more abundant in early- than in late-replicating domains. Surprisingly, after controlling these variables, no difference in ORC/MCM density is detected between initiation zones, termination zones, unidirectionally replicating regions, and randomly replicating regions. Therefore, ORC/MCM density correlates with replication timing but does not solely regulate the probability of replication initiation. Interestingly, H4K20me3, a histone modification proposed to facilitate late origin licensing, was enriched in late-replicating initiation zones and gene deserts of stochastic replication fork direction. We discuss potential mechanisms specifying when and where replication initiates in human cells.

## Introduction

In human cells, DNA replication initiates from 20,000 to 50,000 replication origins selected from a five- to tenfold excess of potential or 'licensed' origins (*Moiseeva and Bakkenist, 2018*; *Papior et al., 2012*). Origin licensing, also called pre-replicative complex (pre-RC) formation, occurs in late mitosis and during the G1 phase of the cell cycle. During this step, the origin recognition complex (ORC) binds DNA and, together with Cdt1 and Cdc6, loads minichromosome maintenance complexes (MCM), the core motor of the replicative helicase, as inactive head-to-head double hexamers (MCM-DHs) around double-stranded DNA (*Bell and Kaguni, 2013*; *Evrin et al., 2009*;

*Remus and Diffley, 2009*). A single ORC reiteratively loads multiple MCM-DHs. However, once MCM-DHs have been assembled, ORC does not maintain contact with the MCM-DH and neither ORC, nor Cdc6, nor Cdt1 are required for origin activation (*Fragkos et al., 2015*; *Hyrien, 2016*; *Powell et al., 2015*; *Remus et al., 2009*; *Rowles et al., 1999*; *Sun et al., 2014*; *Yeeles et al., 2015*). During S phase, CDK2 and CDC7 kinase activities in conjunction with other origin-firing factors convert some MCM-DHs into pairs of active CDC45-MCM-GINS helicases that nucleate bidirectional replisome establishment (*Douglas et al., 2018*; *Moiseeva and Bakkenist, 2018*). MCM-DHs that do not initiate replication are dislodged from DNA during replication.

In *Saccharomyces cerevisiae*, origins are genetically defined by specific DNA sequences (*Marahrens and Stillman, 1992*). In multicellular organisms, no consensus sequence for origin activity has been identified and replication initiates from flexible locations. Although mammalian origins fire at different times through S phase, neighboring origins tend to fire at similar times, partitioning the genome into ~5,000 replication timing domains (RTDs) (*Rivera-Mulia and Gilbert, 2016a*). RTDs replicate in a reproducible order through S phase (*Pope et al., 2014*; *Zhao et al., 2017*). One model for this temporal regulation suggests that RTDs are first selected for initiation, followed by stochastic origin firing within domains (*Boulos et al., 2015*; *Pope et al., 2014*; *Rhind and Gilbert, 2013*; *Rivera-Mulia and Gilbert, 2016b*). A cascade or domino model suggests that replication first initiates at the most efficient (master) origins and then spreads to less efficient origins within an RTD (*Boos and Ferreira, 2019*; *Guilbaud et al., 2011*). Various processes and factors contribute to origin specification such as transcription, DNA sequences, histone variants, histone modifications, and nucleosome dynamics (*Akerman et al., 2020*; *Cayrou et al., 2015*; *Long et al., 2020*; *Petryk et al., 2016*; *Prioleau and MacAlpine, 2016*; *Smith and Aladjem, 2014*). For example, we proposed H4K20me3 to support the licensing of a subset of late-replicating origins in heterochromatin (*Brustel et al., 2017*). Recently, the histone variant H2A.Z has been implicated in ORC recruitment at early origins through deposition of H4K20me2 by histone methyltransferase SUV420H1 (*Long et al., 2020*). Furthermore, binding sites for the origin-firing factor Treslin-MTBP often feature a nucleosome-free gap adjacent to H3K4me2 (*Kumagai and Dunphy, 2020*).

Different approaches have been developed to characterize mammalian origins. Origins have been mapped at the single-molecule level by optical methods or at the cell-population level by sequencing various purified replication intermediates, such as short nascent strands, replication bubbles, and Okazaki fragments (*Hulke et al., 2020*). Strand-oriented sequencing of Okazaki fragments (OK-seq) reveals the population-averaged replication fork direction (RFD) allowing to map initiation and termination (*Chen et al., 2019*; *McGuffee et al., 2013*; *Petryk et al., 2016*; *Smith and Whitehouse, 2012*). Bubble-seq (*Mesner et al., 2013*), single-molecule analyses (*Demczuk et al., 2012*; *Lebofsky et al., 2006*; *Norio et al., 2005*), and OK-seq (*Petryk et al., 2016*; *Wu et al., 2018*) studies of human cells all suggest that replication initiates in broad but circumscribed zones consisting of multiple, individually inefficient sites. OK-seq revealed both, early-firing initiation zones (IZs), which are precisely flanked on one or both sides by actively transcribed genes, and late-firing IZs distantly located from active genes (*Petryk et al., 2016*). Recently, an excellent agreement was observed between early-firing IZs determined by OK-seq and by EdUseq-HU, which identifies nascent DNA synthesized in early S phase cells in the presence of EdU and hydroxyurea (*Tubbs et al., 2018*). Furthermore, high-resolution Repli-seq identified both early and late IZs consistent with OK-seq IZs (*Zhao et al., 2020*).

Chromatin immunoprecipitation followed by sequencing (ChIP-seq) was used to map ORC and MCM chromatin binding. In *Drosophila,* ORC often binds next to open chromatin marks found at transcription start sites (TSSs) (*MacAlpine et al., 2010*), but MCMs, initially loaded next to ORC, are more abundantly loaded and widely redistributed when cyclin E/CDK2 activity rises in late G1 (*Powell et al., 2015*). In human cells, ChIP-seq of single ORC subunits identified from 13,000 to 101,000 ORC potential binding sites (*Dellino et al., 2013*; *Long et al., 2020*; *Miotto et al., 2016*). These studies consistently demonstrated a correlation of ORC-DNA binding with TSSs, open chromatin regions, and early replication timing (RT). ChIP-seq of Mcm7 in HeLa cells suggested that MCM-DHs bind regardless of the chromatin environment, but are preferentially activated upstream of active TSSs (*Sugimoto et al., 2018*). We and others previously used the Epstein–Barr virus (EBV), whose replication in latency is entirely dependent on the human licensing machinery, to compare ORC and MCM binding and replication initiation sites (*Chaudhuri et al., 2001*; *Dhar et al., 2001*; *Papior et al., 2012*; *Ritzi et al., 2003*; *Schepers et al., 2001*). A five- to tenfold excess of potential

origins were licensed per genome with respect to 1–3 mapped initiation event(s) (*Norio, 2001*; *Norio and Schildkraut, 2004*; *Papior et al., 2012*). These findings support the model that human replication initiates in zones, which comprise multiple, individually inefficient sites.

Here, we present the first comparative survey of four different pre-RC components, replication initiation events, transcription activity, and RT in the genome of the human lymphoblastoid Raji cell line by combining ChIP-seq, OK-seq, RNA-seq, and previously published Repli-seq data (*Sima et al., 2018*). We find that, in pre-replicative (G1) chromatin, ORC and MCM are broadly distributed over the genome with high ORC density better correlating with early RT than MCM density. ORC/MCM are depleted from actively transcribed gene bodies and enriched at active gene promoters. ORC/MCM density is homogeneous over non-transcribed genes and intergenic regions of comparable RT. Furthermore, regions of similar RT show a similar ORC/MCM density, be they IZs, termination zones, undirectionally replicating regions (presumably lacking initiation events), or randomly replicating regions. These findings suggest that ORC/MCM densities do not solely determine IZs and that a specific contribution of the local chromatin environment is required. Indeed, we previously showed that IZs are enriched in open chromatin marks typical of active or poised enhancers (*Petryk et al., 2016*). We further show that a subset of non-genic late IZs is enriched in H4K20me3, confirming previous finding that H4K20me3 enhances origin activity in certain chromatin environments (*Brustel et al., 2017*; *Shoaib et al., 2018*).

These findings support the cascade model for replication initiation: the entire genome (except transcribed genes) is licensed for replication initiation. Additional process and factors like adjacent active transcription and epigenetic marks are required to specify master zones of higher replication initiation efficiency. The distributed licensing pattern allows the stochastic activation of secondary origins, possibly triggered by approaching replication forks.

## Results

### Moderate averaging is a suitable approach for ORC and MCM-DH distribution analysis

We used centrifugal elutriation to obtain a G1-enriched, pre-replicative population of human lymphoblastoid Raji cells (*Papior et al., 2012*). Propidium iodide staining followed by FACS (*Figure 1—figure supplement 1a*) and western blot analyses of cyclins A, B, and H3S10 phosphorylation (*Figure 1—figure supplement 1b*) confirmed the cell cycle stages of elutriated fractions. To ensure unbiased mapping of ORC and MCM, we simultaneously targeted Orc2, Orc3, Mcm3, and Mcm7 using validated ChIP-grade antibodies (validated in *Papior et al., 2012*; *Ritzi et al., 2003*; *Schepers et al., 2001*). ChIP efficiencies and qualities were measured using the EBV latent origin *oriP* as reference (*Figure 1—figure supplement 1c*). Raji cells contain 50–60 EBV episomes, allowing an easy detection of ORC/MCM at *oriP* (*Adams et al., 1973*). The viral protein EBNA1 recruits ORC to *oriP's* dyad symmetry element, followed by MCM-DH loading. We detected both ORC and MCM at the dyad symmetry element in G1, whereas a population containing S-G2-M-phased cells depict a reduction in MCM levels, as expected (*Figure 1—figure supplement 1c*; *Papior et al., 2012*; *Ritzi et al., 2003*).

ChIP-seq of two replicates for ORC subunits (Orc2, Orc3) and of three replicates for MCM proteins (Mcm3, Mcm7) resulted in reproducible, but dispersed, ChIP-seq signals as exemplified by the well-characterized replication origin Mcm4/PRKDC (*Figure 1a*; *Ladenburger et al., 2002*; *Schaarschmidt et al., 2002*). We employed the MACS2 peak-calling program (*Feng et al., 2012*; *Zhang et al., 2008*), but found that the obtained results were too dependent on the chosen program settings and that ORC and MCM distributions were too dispersed to be efficiently captured by peak calling (data not shown), requiring an alternative approach.

Consequently, we summed up the reads of the ChIP replicates at different binning sizes and normalized the signals against the mean read frequencies of each ChIP sample and against input, as is standard in most ChIP-seq analyses. We computed the Pearson correlation coefficients between ORC/MCM ChIPs and obtained good correlations at 1 kb bin size and only marginal improvement at larger sizes (*Figure 1—figure supplement 2a*). When working in 1 kb bins, we still detected the enrichment of ORC/MCM at the MCM/PRKDC origin (*Figure 1b*), indicating that we do not lose local, biologically relevant signals.

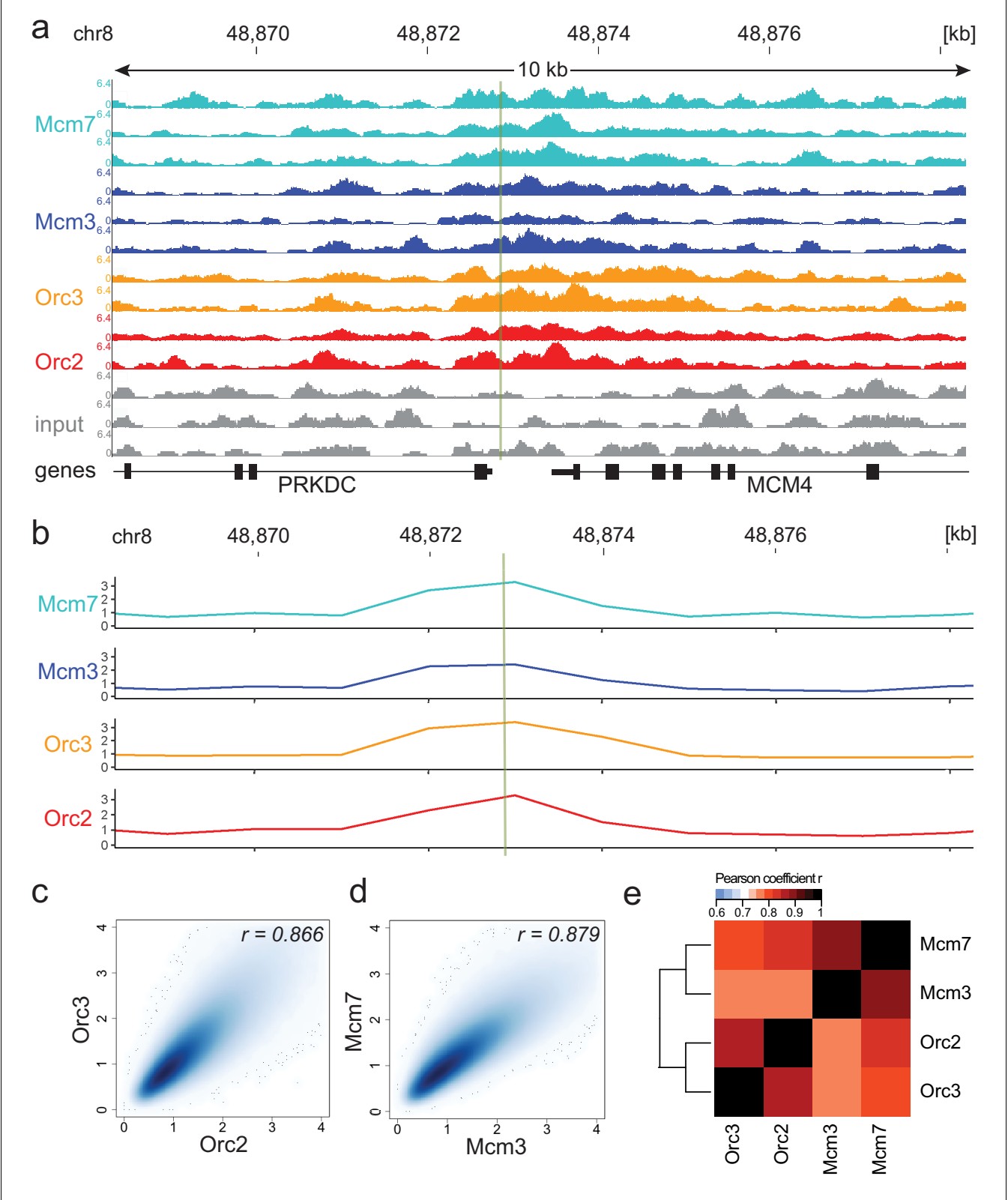

**Figure 1.** Moderate averaging represents a valid approach for origin recognition complex/minichromosome maintenance complex (ORC/MCM) chromatin immunoprecipitation followed by sequencing (ChIP-seq) analysis. (a) Sequencing profile visualization in UCSC Genome Browser (hg19) at the Mcm4/PRKDC origin after reads per genomic content normalization: two samples of Orc2 and Orc3, and three samples of Mcm3 and Mcm7, are plotted against the input in three replicates. The profiles are shown in a 10 kb window (chr8: 48,868,314–48,878,313); the mapped position of the

*Figure 1 continued on next page*

*Figure 1 continued*

origin is indicated as green line. (**b**) The profile of ORC/MCM ChIP-seq after 1 kb binning at the same locus. The reads of replicates were summed and normalized by the total genome-wide ChIP read frequency followed by input division. Y-axis represents the resulting relative read frequency. (**c**) Correlation plot between Orc2 and Orc3 relative read frequencies in 1 kb bins. (**d**) Correlation plot between Mcm3 and Mcm7 relative read frequencies in 1 kb bins. (**e**) Heatmap of Pearson correlation coefficients r between all ChIP relative read frequencies in 1 kb bins. Column and line order were determined by complete linkage hierarchical clustering using the correlation distance (d = 1 r). Refer to *Figure 1—figure supplement 3* for data representation without input division.

The online version of this article includes the following figure supplement(s) for figure 1:

**Figure supplement 1.** Experimental validation of cell cycle fractionation and origin recognition complex and minichromosome maintenance complex (ORC/MCM) chromatin immunoprecipitation followed by sequencing quality.

**Figure supplement 2.** The input sequencing control is differentially represented in regions of biological function.

**Figure supplement 3.** Origin recognition complex/minichromosome maintenance complex (ORC/MCM) enrichments at the MCM4/PRKDC origin persists without input normalization.

In line with a previous report (*Teytelman et al., 2009*), the input control was significantly under-represented in DNase hypersensitive (HS) regions, at TSSs, and at early RTDs (*Figure 1—figure supplement 2b–d*). As sonication-hypersensitive regions correlate with DNase HS regions (*Schwartz et al., 2005*), we carefully compared our results obtained with and without input normalization. For example, we still detect enrichment of ORC/MCM at the MCM/PRKDC origin when we omit input normalization (*Figure 1—figure supplement 3*). As will become apparent, similar conclusions were obtained in further analyses performed with or without input normalization.

The reliability and reproducibility of our ChIP experiments is reflected by the high Pearson correlation coefficients of the relative read frequencies of Orc2/Orc3 (r = 0.866, *Figure 1c*) and Mcm3/Mcm7 (r = 0.879, *Figure 1d*). The correlations between ORC and MCM were only slightly lower (Mcm3/Orc2/3: r = 0.775/0.757; Mcm7/Orc2/3: r = 0.821/0.800, *Figure 1e*). Hierarchical clustering based on Pearson correlation of ChIP profiles clustered ORC and MCM profiles together. Similar results were obtained using non input-normalized data (*Figure 1—figure supplement 3b–d*). To compare our ChIP-seq data to previously published Orc2 ChIP-seq from asynchronously cycling K562 cells (GSE70165; *Miotto et al., 2016*), we calculated the relative read frequencies of our ORC ChIPs around an aggregate of K562 Orc2 peaks (>1 kb) and found substantial enrichment (*Figure 1—figure supplement 3e*). *Miotto et al., 2016* reported that Orc2 co-localizes with DNase HS sites present at active promoters and enhancers. In line with these observations, we found a significant enrichment of ORC at DNase HS regions > 1 kb, compared to regions deprived of DNase HS sites, with or without input normalization (*Figure 1—figure supplement 3f, g*). These results further validate our data.

## ORC/MCM are enriched in IZs dependent on transcription

We next compared the relative read frequencies of ORC/MCM to active replication initiation units. Using OK-seq in Raji cells (*Wu et al., 2018*), we calculated the RFD (see Materials and Methods) and delineated preferential replication IZs as ascending segments (ASs) of the RFD profile. RFD profiles present upshifts that define origins to kilobase resolution in yeast (*McGuffee et al., 2013*), but in mammalian cells these transitions are more gradual, extending over 10–100 kb (*Chen et al., 2019*; *McGuffee et al., 2013*; *Petryk et al., 2016*; *Tubbs et al., 2018*; *Wu et al., 2018*). We analyzed ASs > 20 kb, allowing to assess ChIP signals up to 10 kb within ASs (see Materials and Methods). Using the RFD shift across the ASs (ΔRFD) as a measure of replication initiation efficiency, we further required ΔRFD > 0.5 to select the most efficient IZs. In total, we selected 2957 ASs, with an average size of 52.3 kb, which covered 4.9% (155 Mb) of the genome (*Figure 2a*, green bars, *Table 1*). In total, 2451 (83%) of all AS located close to genic regions (ASs extended by 20 kb on both sides overlapped with at least one annotated gene). Performing RNA-seq in asynchronously cycling Raji cells, we determined that 673 ASs (22.8% of all ASs) were flanked by actively transcribed genes (transcripts per kilobase per million [TPM] >3) on both sides (type 1 AS), with less than 20 kb between AS borders and the closest transcribed gene. In total, 1026 ASs (34.7%) had only one border associated to a transcribed gene (type 2 AS; TPM >3). Also, 506 ASs (17.1%) were devoid of proximal genes (non-genic AS) (*Table 1*). The slope did not change considerably between the different AS types, although type 1 ASs were on average slightly more efficient, followed by type 2 ASs,

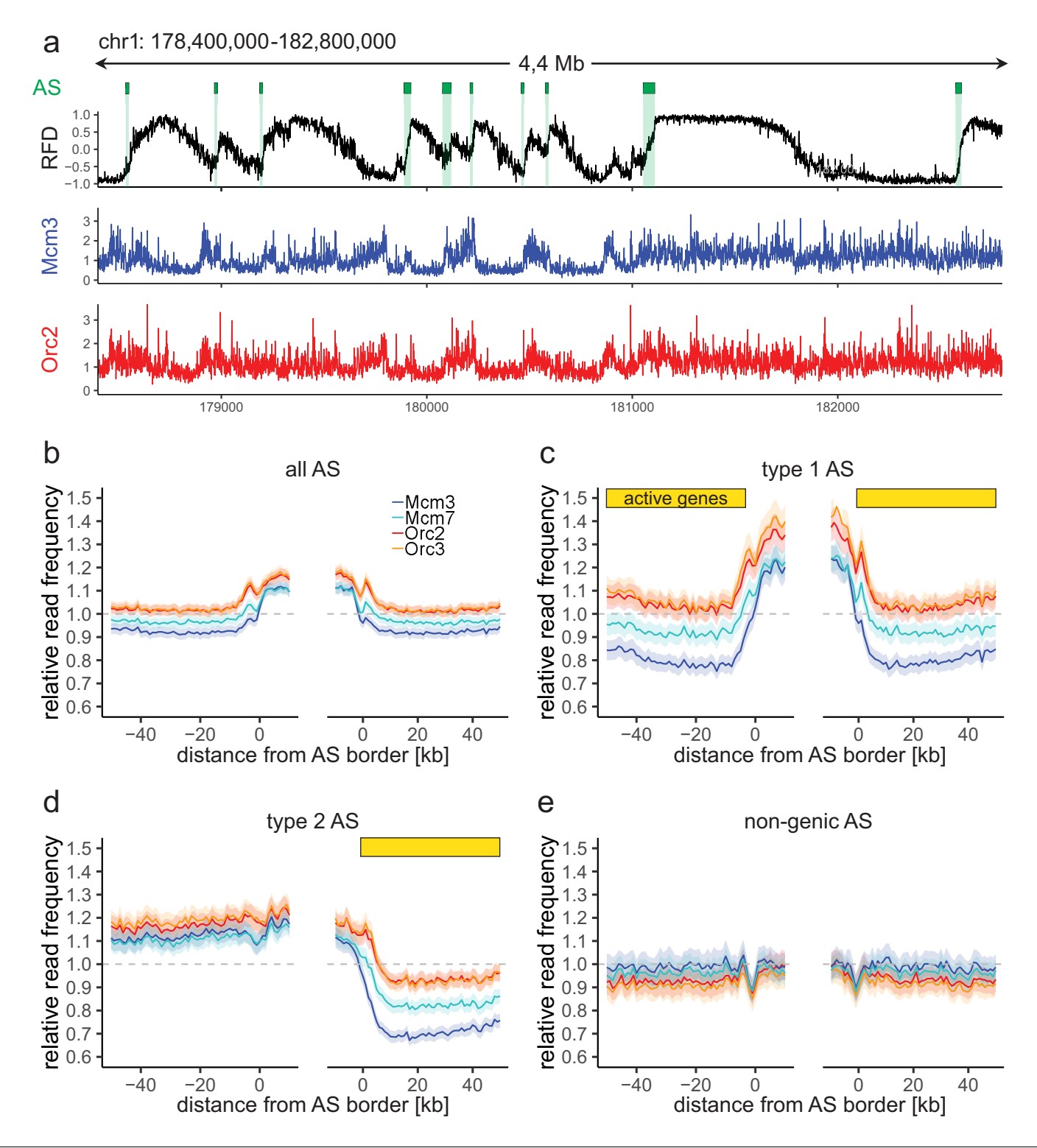

**Figure 2.** Origin recognition complex/minichromosome maintenance complex (ORC/MCM) enrichment within ascending segments (ASs) depends on active transcription. (**a**) Top panel: example of an replication fork direction (RFD) profile on chr1: 178,400,000–182,800,000, covering 4 Mb. Detected ASs are labeled by green rectangles (irrespective of length and RFD shift). Middle and bottom panels: representative Mcm3 (blue) and Orc2 (red) chromatin immunoprecipitation followed by sequencing (ChIP-seq) profiles after binning for the same region. (**b–e**) Average input-normalized relative ChIP read frequencies of Orc2, Orc3, Mcm3, and Mcm7 at AS borders of (**b**) all AS (L > 20 kb and ΔRFD >0.5; n = 2957), (**c**) type 1 ASs with transcribed genes at

*Figure 2 continued on next page*

*Figure 2 continued*

both AS borders (n = 673), (**d**) type 2 ASs oriented with their AS border associated to transcribed genes at the right (n = 1026), and (**e**) non-genic ASs in gene-deprived regions (n = 506). The mean of ORC and MCM relative read frequencies is shown ±2 × SEM (lighter shadows). The dashed grey horizontal line indicates relative read frequency 1.0 for reference. For type 1 and 2 ASs, yellow bars mark the AS borders associated to transcribed genes. Refer to *Figure 2—figure supplement 2* for analysis without input division.

The online version of this article includes the following figure supplement(s) for figure 2:

**Figure supplement 1.** Characterization of different ascending segment (AS) types.

**Figure supplement 2.** Origin recognition complex/minichromosome maintenance complex (ORC/MCM) enrichments within ascending segments (ASs) without input normalization.

then non-genic ASs (*Figure 2—figure supplement 1a*). Type 1 and 2 ASs located within early RTDs, while non-genic ASs were predominantly late replicating (*Figure 2—figure supplement 1b*), as previously observed in GM06990 and HeLa (*Petryk et al., 2016*).

To study the relationship between ORC/MCM densities and replication initiation, we computed the relative read frequencies of ORC/MCM around all AS aggregate borders. Both ORC and MCM were, on average, enriched within ASs compared to flanking regions (*Figure 2b*, *Figure 2—figure supplement 2a* without input division). To resolve the impact of transcriptional activity, we repeated this calculation for the different AS types (*Figure 2c-e*; non-input-normalized data in *Figure 2—figure supplement 2b–d*). Transcriptional activity in AS flanking regions was associated with increased ORC/MCM levels inside ASs (compare *Figure 2b, c*) and a prominent MCM depletion from transcribed regions (*Figure 2c, d*, right border). In contrast, in type 2 ASs, ORC/MCM levels remained elevated at non-transcribed AS borders (*Figure 2d*, left border). No ORC/MCM enrichment was detected within non-genic ASs (*Figure 2e*).

AS borders associated with transcriptional activity were locally enriched in ORC/MCM (*Figure 2c, d*, both and right borders respectively). This is in line with previously detected Orc1 accumulation at AS borders (*Petryk et al., 2016*). Reciprocally, non-genic AS borders only showed a local dip in ORC/MCM levels (*Figure 2d*, left border; Figure 2e, both borders), but the biological significance of this observation remains unclear. A sequence analysis revealed biased distributions of homopolymeric repeat sequences at AS borders (data not shown). Such sequences may affect nucleosome formation and ORC binding, but may also bias Okazaki fragment/AS border detection at small scales (*Figure 2—figure supplement 1a*) as well as mappability.

## ORC and MCM are depleted from transcribed gene bodies and enriched at TSSs

Consistent with previous OK-seq studies (*Chen et al., 2019*; *Petryk et al., 2016*), the average RFD profile of active genes revealed strong ASs upstream of TSSs and downstream of transcriptional termination site (TTSs), and descending RFD segments (DSs) across the active gene bodies (*Figure 3—figure supplement 1a*). This behavior depended on transcriptional activity as silent genes displayed an overall flat RFD profile (*Figure 3—figure supplement 1a*). When setting our ORC/MCM ChIP-seq data in relation to transcription, we observed that the ORC relative read distribution was significantly

**Table 1.** Characterization of different AS subtypes.

| | Number | Genome coverage (%) | Average length (kb) |
|---|---|---|---|
| All AS | 2957 | 4.9 | 52.3 |
| Genic AS | 2451 | 4.1 | 52.3 |
| Type 1 AS | 673 | 1.1 | 50.7 |
| Type 2 AS | 1026 | 5.2 | 50.2 |
| Non-genic AS | 506 | 0.8 | 50.7 |

Only AS ≥20 kb with ΔRFD > 0.5 were considered.

Genic ASs: ASs extended 20 kb on both sides is overlapped by genic region(s) irrespective of transcriptional activity; type 1 and type 2 AS: ASs flanked by expressed genes (TPM ≥3) within 20 kb on both sides (type 1) or one side (type 2); non-genic: no annotated gene ±20 kb of AS borders; AS: ascending segment; RFD: replication fork direction; TPM: transcripts per kilobase per million.

enriched at active TSSs, as already demonstrated in *Drosophila* (*MacAlpine et al., 2010*) and human cells (*Dellino et al., 2013*; *Miotto et al., 2016*). Thereby, ORC relative read distribution was moderately but significantly higher upstream of TSSs and downstream of TTSs than within active genes (*Figure 3a*). These observations were independent of input normalization (compare *Figure 3a* with *Figure 3—figure supplement 1b*). The depletion of ORC from gene bodies was statistically significant for approximately 45% of actively transcribed genes (*Table 1*). Compared to ORC, Mcm3 and Mcm7 enrichments at TSSs were less prominent, but depletions from gene bodies were more pronounced (*Figure 3a*, *Figure 3—figure supplement 1b*), with 75% and 58% of investigated transcribed gene bodies significantly depleted from Mcm3 and Mcm7, respectively (*Table 1*). Depletion

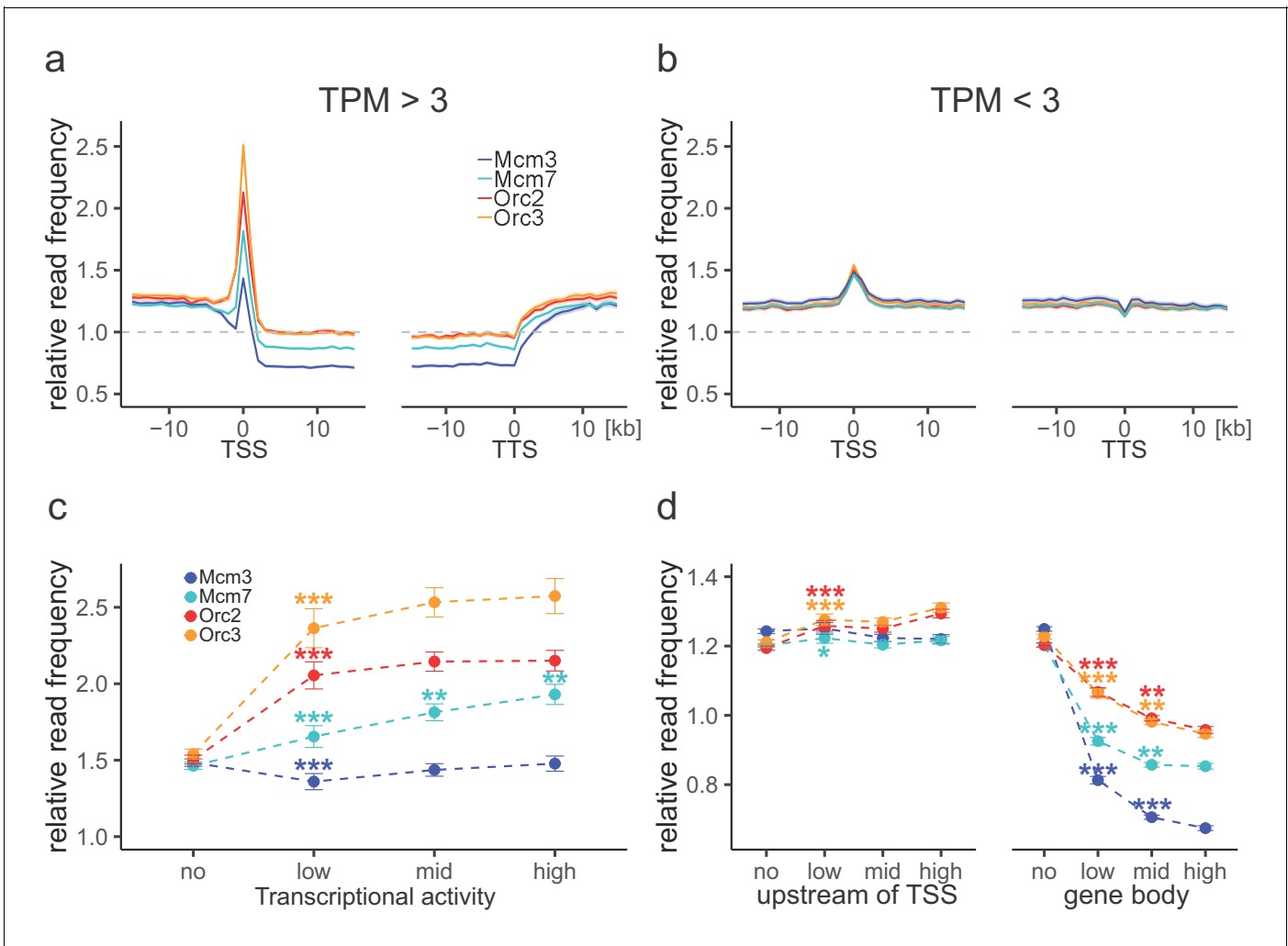

**Figure 3.** Origin recognition complex (ORC) is enriched at active transcription start sites (TSSs) while minichromosome maintenance complex (MCM) is depleted from actively transcribed genes. (a, b) ORC/MCM relative read frequencies around TSSs or transcriptional termination sites (TTSs) for (a) active genes (transcripts per kilobase per million [TPM] >3) and (b) inactive genes (TPM <3). Only genes larger than 30 kb without any adjacent gene within 15 kb were considered. Distances from TSSs or TTSs are indicated in kb. Means of ORC and MCM frequencies are shown ±2 × SEM (lighter shadows). The dashed grey horizontal line indicates relative read frequency 1.0 for reference. (c) ORC/MCM relative read frequencies at TSSs dependent on transcriptional activity (±2 × SEM). (d) ORC/MCM relative read frequencies upstream of TSSs and within the gene body dependent on transcriptional activity (±2 × SEM; TSSs ± 3 kb removed from analysis). Transcriptional activity was classified as no (TPM <3), low (TPM 3–10), mid (TPM 10–40), and high (TPM >40). Statistics were performed by one-way ANOVA followed by Tukey's post-hoc test. p-Values are indicated always comparing to the previous transcriptional level. *p<0.05, **p<0.01, ***p<0.001. Refer to *Figure 3—figure supplement 1* for analyses without input division.

The online version of this article includes the following figure supplement(s) for figure 3:

**Figure supplement 1.** Replication fork direction (RFD) and origin recognition complex/minichromosome maintenance complex (ORC/MCM) profiles without input normalization at gene extremities.

was strictly homogeneous from TSS to TTS, strongly suggesting that transcription itself displaces ORC and MCM-DH complexes (*Figure 3a*). In contrast, at silent genes, ORC/MCM were hardly enriched at TSSs and were not depleted from gene bodies (*Figure 3b*, *Figure 3—figure supplement 1c*). Increasing transcriptional activity (classified as low: 3–10 TPM; mid: 10–40 TPM; high:>40 TPM) did not have any major impact on ORC/MCM enrichments at TSSs (*Figure 3c*, *Figure 3—figure supplement 1d*). ORC/MCM depletion within gene bodies was slightly more pronounced with increasing transcription levels when normalized for input (*Figure 3d*), but this was less convincing without input normalization (*Figure 3—figure supplement 1e*). Basal ORC/MCM levels upstream of TSSs and downstream of TTSs were identical, indicating that the local ORC enrichment at TSSs did not result in more MCM loading upstream than downstream of active genes.

Our observation that Mcm3 and Mcm7 are significantly depleted from transcribed gene bodies is consistent with their active displacement by transcription in G1, as previously proposed in *Drosophila* (*Powell et al., 2015*) and human cells (*Macheret and Halazonetis, 2018*). This depletion process contributes to delineating IZs flanked by active genes. In contrast, ORC/MCM density remained constant across non-genic AS borders (*Figure 2*), suggesting that ORC/MCM are not sufficient to delimit non-genic replication IZs.

## ORC/MCM genomic distributions are broad and correlate with RT but not IZs

RT is a crucial aspect of genome stability that is correlated with gene expression and chromatin structure, which coordinate the selection of origins and timing of origin firing (*Knott et al., 2009*). In yeast, it has been reported that the number of MCM-DHs loaded at origins correlates with RT, suggesting how RT profiles can emerge from stochastic origin firing (*Das et al., 2015*; *Yang et al., 2010*). In human cells, ORC binding data have also been used to predict RT profiles (*Miotto et al., 2016*). To clarify the relationships between IZ location, IZ firing time, and ORC/MCM density in human cells, we used Raji Early/Late Repli-seq data from *Sima et al., 2018* and related RT to ORC/MCM relative read frequencies and RFD slope (*Sima et al., 2018*).

We analyzed four different types of RFD pattern categories (exemplified in *Figure 4—figure supplement 1a, b*) as previously defined in *Petryk et al., 2016*: (i) ascending RFD segments (ASs), that is, predominant-IZs; (ii) descending RFD segments (DSs), that is, predominant-termination zones (TZ); (iii) flat segments of high |RFD| (|RFD| > 0.8 over >300 kb), that is, unidirectionally replicating regions (URRs), where replication forks always migrate in the same direction, implying a lack of initiation events; and (iv) flat segments of null RFD regions (NRRs; |RFD| < 0.15 over >500 kb), presumably replicating by random initiation and termination, mainly observed in late-replicating gene deserts (*Figure 4—figure supplement 1c*).

We calculated relative Orc2 and Mcm3 (*Figure 4*, *Figure 4—figure supplement 1d, e* for Orc3 and Mcm7) read frequencies in 10 kb bins against RT in intergenic regions (left column), silent gene bodies (TPM <3, middle column), or active gene bodies (TPM >3, right column). We considered either all bins (top row) or bins corresponding to ASs, DSs, URRs, and NRRs (following rows in descending order). Histograms were normalized by column, that is, each column is the probability density function of ChIP frequency at a given RT bin.

Consistently with *Figure 3a, b*, expressed genes showed lower ORC/MCM densities than silent genes and intergenic regions (*Figure 4*, *Figure 4—figure supplement 1d, e*). This was particularly significant in early- and mid-replicating regions, as demonstrated by Kolmogorov–Smirnov statistics (*Figure 4—figure supplement 2a*, red circles). The depletion was more pronounced for MCM than ORC, as already noted in *Figure 3a*. In contrast, the difference between silent genes and intergenic regions was at best marginally significant (blue circles). In all cases, ORC/MCM densities monotonously decreased from early to late RT windows, but this RT dependency was much attenuated in expressed genes, particularly for MCM, as expected if transcription removes this complex from both early- and late-replicating genes (*Figure 4b*, *Figure 4—figure supplement 1e*).

Strikingly, our analysis did not reveal any clear differences in the levels of ORC/MCM between intergenic ASs, DSs, URRs, and NRRs when bins of similar RT were compared (*Figure 4—figure supplement 2a*). A similar behavior was also apparent for ASs, DSs, URRs, and NRRs in silent genes and for DSs and URRs in active genes. Note that the few (579) bins corresponding to ASs in active genes are probably misleading as they are mainly attributable to annotation errors: the annotated gene overlapped the AS but the RNA-seq signal was confined outside the AS (*Figure 4*; *Figure 4—figure*

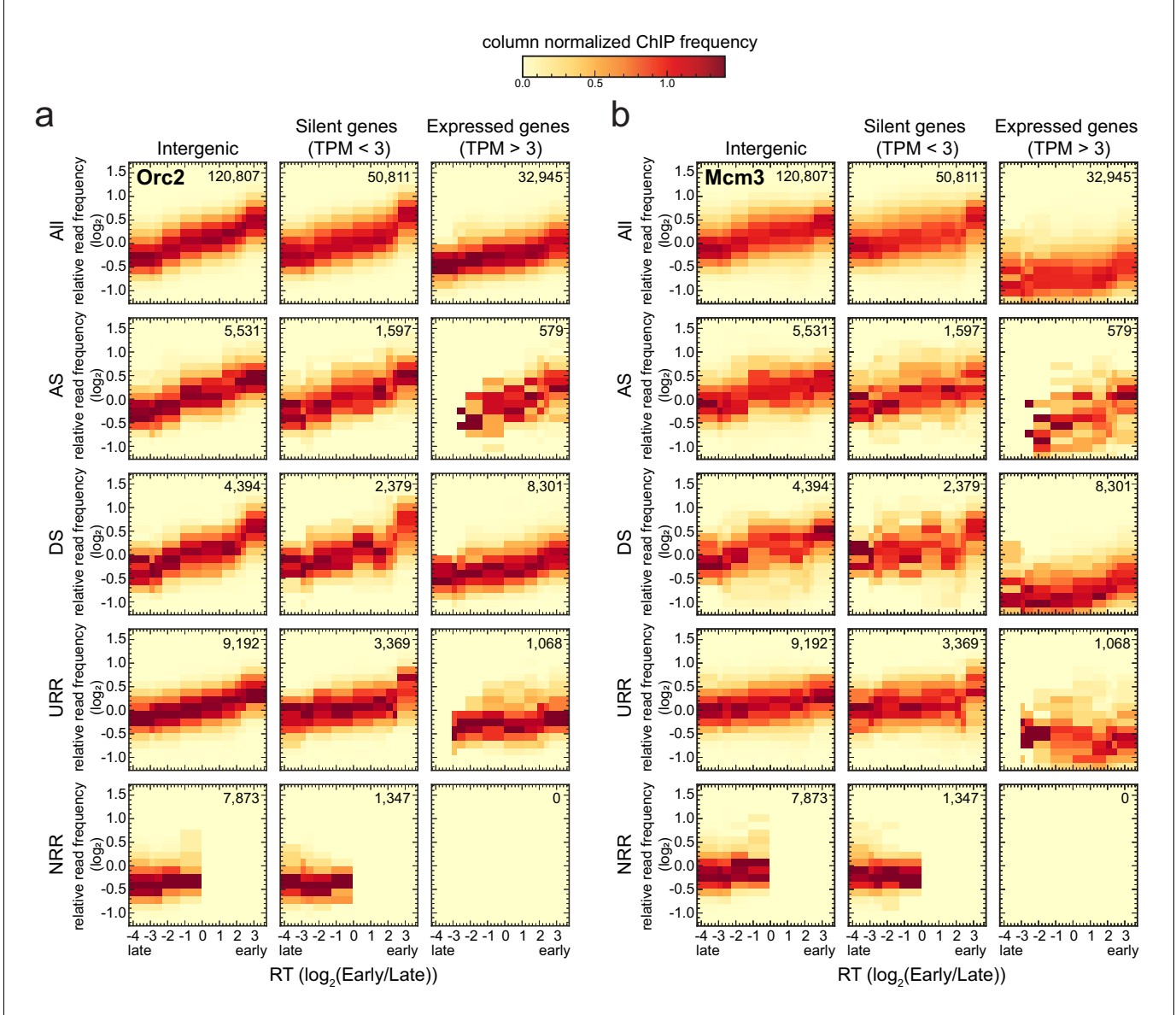

**Figure 4.** Origin recognition complex/minichromosome maintenance complex (ORC/MCM) levels correlate with replication timing (RT) and transcriptional activity but are otherwise homogeneously distributed along the genome and uncorrelated to replication fork direction (RFD) patterns. (a, b) 3 × 5 panel of 2D histograms of Orc2 (a) and Mcm3 (b) chromatin immunoprecipitation (ChIP) relative read frequency versus RT (average log$_2$(Early/Late) over 100 kb binned according to the decile of RT distribution). The analysis was performed in 10 kb bins. Histograms are normalized by column and represent the probability density functions of ChIP relative frequencies at a given replication timing. The color legend is indicated on top. 2D histograms are computed for intergenic regions (left column), silent genes (transcripts per kilobase per million TPM <3, middle column), and expressed genes (TPM >3, right column). Transcription start sites and transcriptional termination sites proximal regions were not considered (see Materials and methods). The rows show either all bins (top row) or restriction to ascending segment (AS) bins (predominant replication initiation, second row), descending segment (DS) bins (descending segment, predominant replication termination, third row), unidirectionally replicating region (URR) bins (unidirectional replication, no initiation, no termination, fourth row), and null RFD region (NRR) bins (null RFD regions, spatially random initiation and termination, bottom row). The number of bins per histogram is indicated in each panel. See *Figure 4—figure supplement 1* for equivalent Orc3 and Mcm7 analyses. Refer to *Figure 4—figure supplement 2a* for statistical comparisons.

The online version of this article includes the following figure supplement(s) for figure 4:

**Figure supplement 1.** Origin recognition complex/minichromosome maintenance complex (ORC/MCM) levels are correlated with replication timing (RT) and transcriptional activity but otherwise homogeneously distributed along the genome and uncorrelated to replication fork direction (RFD) patterns.

**Figure supplement 2.** Kolmogorov–Smirnov statistics between the origin recognition complex/minichromosome maintenance complex (ORC/MCM).

*supplement 1d, e*; *Petryk et al., 2016*). In summary, the densities of ORC/MCM across genomic segments were related to RT and gene expression but were not predictive of any RFD pattern.

Strictly speaking, the slope of an RFD segment is proportional to the difference between the density of initiation and termination events within the segment (*Audit et al., 2013*). Therefore, we cannot exclude delocalized initiation events within DSs, which would explain why DSs were not significantly depleted in ORC/MCM compared to ASs (*Figure 4—figure supplement 2a*). In contrast, we can almost certainly exclude initiation events within URRs, but their ORC/MCM densities were not significantly lower than in ASs. This suggests that specific mechanisms repress potential origins in URRs and/or activate them in ASs.

Taken together, these results suggest that the density of ORC/MCM is not a reliable predictor of initiation probability, even though ORC density (and to a lesser extent MCM density) well correlated with RT. Thus, potential origins are widespread through the genome, but additional genetic or epigenetic factors are regulating whether and when they fire.

## Cell cycle dynamics of ORC and MCM binding

The results above revealed a gradient of ORC/MCM densities according to RT. To confirm this observation, we extracted early and late RTDs employing a threshold of $\log_2(\text{Early/Late}) > 1.6$ for early RTDs and $<-2.0$ for late RTDs, which resulted in 302 early RTDs covering 642.8 Mb and 287 late RTDs covering 617.4 Mb of the genome. Restricting the analysis to intergenic regions, we calculated the mean ORC/MCM relative read frequencies of pre-replicative G1-phased chromatin in early compared to late RTDs. ORC was 1.4 times enriched in early RTDs compared to late RTDs (*Figure 5a*, *Figure 4—figure supplement 2b*, *Table 2*). Mcm3 and Mcm7 levels, although showing the same tendencies, were less contrasted than ORC.

To confirm the biological relevance of this finding, we repeated this analysis using chromatin from late S-G2-M chromatin (elutriation fraction 80 ml/min; *Figure 1—figure supplement 1a, b*), when replication has displaced most of the MCMs, as exemplified by qPCR at *oriP's* dyad symmetry element (*Figure 1—figure supplement 1c*). In S-G2-M chromatin, ORC was still significantly enriched in early RTDs, compared to late RTDs, but MCMs were not, consistent with completed replication of early but not late RTDs (*Figure 5b*, *Figure 4—figure supplement 2c*, *Table 2*). These results demonstrate that the MCM signal is dynamic through the cell cycle as expected. These results also show that the preferential binding of ORC to early replicating (open) chromatin is not dependent on cell cycle stage.

Given that Orc1 in human cells is degraded at the G1-S transition and in early S phase (*Kreitz et al., 2001*; *Méndez et al., 2002*; *Ohta et al., 2003*), it might appear surprising that we detect Orc2 and Orc3 binding to S-G2-M chromatin. ChIP-seq only allows monitoring the relative distribution of chromatin-bound proteins along the genome and not their absolute levels. We therefore do not exclude that Orc2 and Orc3 binding to chromatin is globally and origin-specifically decreased after G1-S entry (*Gerhardt et al., 2006*; *Siddiqui and Stillman, 2007*). In human cells, Orc1 reappears as cells enter mitosis and is the first ORC subunit to bind to mitotic chromosomes, but other ORC subunits seem to join only in daughter G1 cells (*Kara et al., 2015*). Nevertheless, GFP-tagged Orc1 was found to associate with chromatin throughout mitosis in living Chinese hamster cells and to co-localize with Orc4 in metaphase spreads (*Okuno et al., 2001*). The binding of Orc2 and Orc3 we detect in S-G2-M may either occur independently of Orc1 or reflect the binding of the entire complex in late mitotic cells.

## Late-replicating non-genic ASs and NRRs are characterized by H4K20me3

We and others recently demonstrated that H4K20me3 is involved in licensing a subset of late-replicating regions (*Benetti et al., 2007*; *Brustel et al., 2017*; *Pannetier et al., 2008*). Here, we looked further into the relation between this chromatin mark, ORC/MCM, and replication initiation. We performed ChIP for H4K20me3 and H4K20me1 in three replicates from G1-phased cells and validated them by qPCR (*Figure 5—figure supplement 1a, b*). An exemplary H4K20me3 profile is shown along ORC/MCM profiles in *Figure 5—figure supplement 1d*. We performed MACS2 broad peak detection, keeping only peaks overlapping in all three samples (16,852 peaks for H4K20me3 and 12,264 peaks for H4K20me1, ranging in size from 200 bp to 105 kb and 183 kb, respectively;

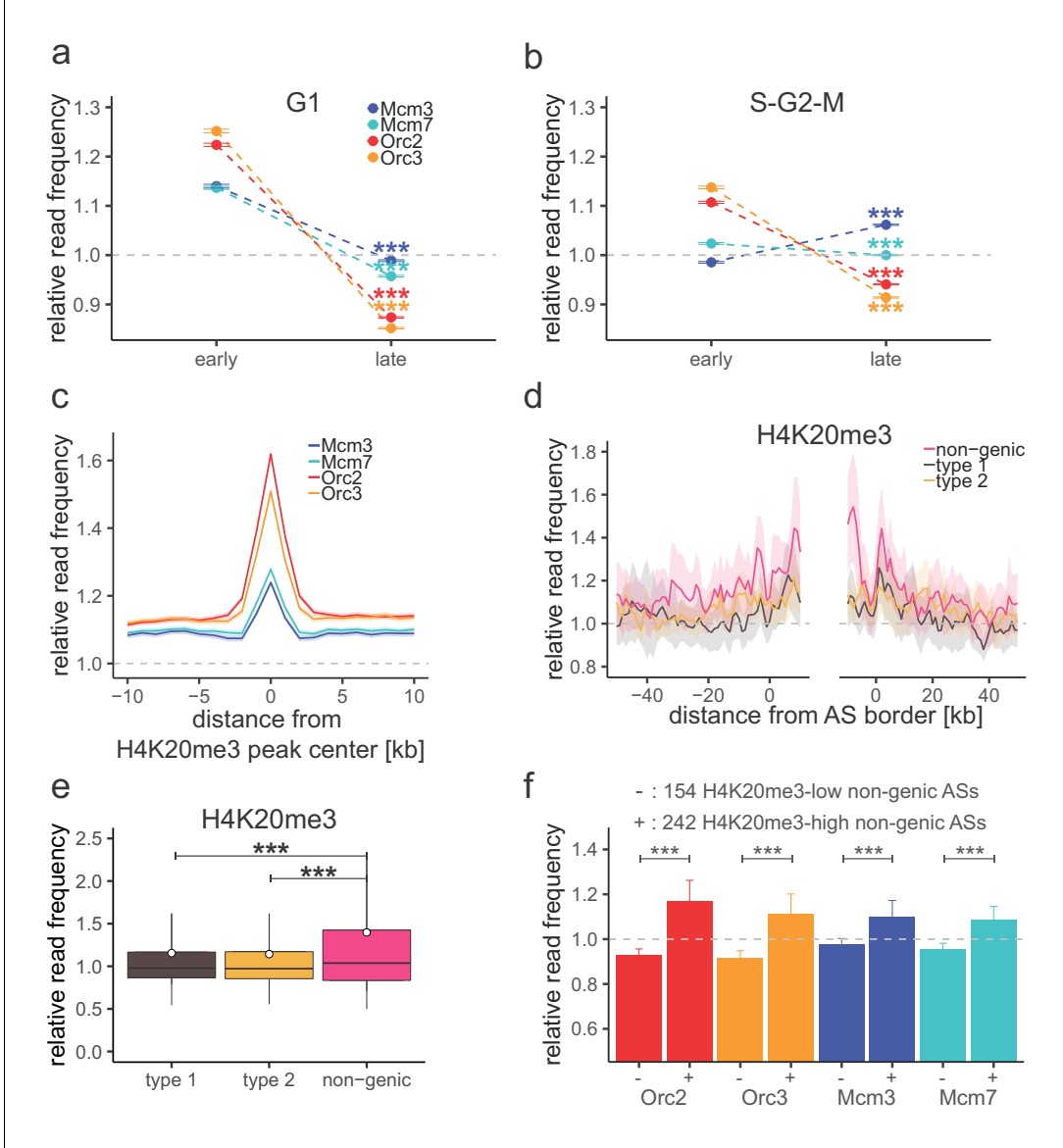

**Figure 5.** H4K20me3 selectively marks a subset of late-replicating non-genic ascending segments (ASs). (a) Origin recognition complex/minichromosome maintenance complex (ORC/MCM) G1 chromatin relative read frequencies (±2 × standard error of the mean [SEM]) in early or late replication timing domains (RTDs). Early RTDs were defined as $\log_2$(Early/Late) > 1.6; late RTDs < −2.0. The analysis was performed in 10 kb bins. Any gene ±10 kb was removed from the analysis. Statistics were performed using one-sided *t*-test. (b) ORC/MCM relative read frequencies (±2 × SEM) obtained from S-G2-M chromatin in early or late RTDs using the same settings as in (a). (c) Average ORC/MCM relative read frequencies at H4K20me3 peaks (>1 kb). (d) H4K20me3 relative read frequencies at AS borders of the different AS types. Type 2 ASs are oriented with their AS borders associated to transcribed genes at the right. Means of H4K20me3 relative read frequencies are shown ±2 × SEM (lighter shadows). (e) Boxplot representation of H4K20me3 relative read frequencies within the different AS types. Boxplot represents the mean (circle), median (thick line), first and third quartile (box), and first and ninth decile (whiskers) of the relative read frequencies in each AS type. Statistics were performed by one-way ANOVA followed by Tukey's post-hoc test. (f) Histogram representation of mean ±2 × SEM of ORC/MCM relative read frequencies in G1 at 242 H4K20me3-low non-genic ASs and 154 H4K20me3-high non-genic ASs. Statistics were performed using one-sided *t*-test. ***p<0.001. Refer to *Figure 5—figure supplement 1* for validation of H4K20me3 chromatin immunoprecipitation.

The online version of this article includes the following figure supplement(s) for figure 5:

**Figure supplement 1.** Origin recognition complex/minichromosome maintenance complex (ORC/MCM) is enriched in late-replicating, H4K20me3-high non-genic ascending segment (AS) and null RFD region (NRR) windows.

**Table 2.** Ratio of chromatin immunoprecipitation (ChIP) mean relative read frequencies in early versus late replication timing domains and G1 versus S-G2-M samples.

| | Mean relative read frequency ratio (early/late) | | Mean relative read frequency ratio (G1/S-G2-M) | |
|---|---|---|---|---|
| | G1 | S-G2-M | Early | Late |
| Orc2 | 1.40 | 1.18 | 1.11 | 0.93 |
| Orc3 | 1.47 | 1.24 | 1.10 | 0.93 |
| Mcm3 | 1.15 | 0.93 | 1.16 | 0.93 |
| Mcm7 | 1.19 | 1.02 | 1.11 | 0.96 |

Calculated in 10 kb bins. All annotated genic regions were removed ± 10 kb.

*Table 2*, *Figure 5—figure supplement 1c*). We calculated ORC/MCM relative read frequencies binned at 1 kb resolution at H4K20me3/H4K20me1 peaks > 1 kb (12,251 and 6277 peaks, respectively). ORC and, to a lower extent, MCM were enriched at H4K20me3, but not H4K20me1 peaks (*Figure 5c*, *Figure 5—figure supplement 1e*).

H4K20me3 coverage at the different AS types depicts an increased H4K20me3 signal only in non-genic ASs, disclosing the first histone modification specific for late-replicating non-genic ASs (*Figure 5d, e*). Starting from 506 non-genic ASs, we extracted two subsets of 154 and 242 non-genic ASs, where H4K20me3 relative read frequencies were above the mean genome value by more than 1.5× standard deviation, or below the genome mean value, respectively. ORC/MCM were enriched at the H4K20me3-high subgroup compared to the H4K20me3-low subgroup (*Figure 5f*). These results suggest that the presence of H4K20me3 at transcriptionally independent, non-genic ASs may contribute at the origin-licensing step to specifying these regions as highly efficient 'master' initiation zones (Ma-IZs) in late-replicating DNA. The difference of ORC/MCM densities between H4K20me3-high and -low subgroups was less pronounced in chromatin derived from S-G2-M-phased cells. The dynamic differences between the cell cycle fractions confirm the biological relevance of this finding (*Figure 5—figure supplement 1f*).

To further explore the links between H4K20me3 and replication initiation, we analyzed the density of this modification in genome segments of various RT, gene activity, and RFD patterns (*Figure 6a*). Several interesting observations emerged from this analysis. First, the H4K20me3 level was weakly but systematically more abundant in early than in late-replicating chromatin, suggesting that H4K20me3 is not exclusively present in late-replicating heterochromatin. Second, the H4K20me3 level was slightly lower in transcribed genes than in the non-transcribed rest of the genome (*Figure 6a, Figure 5—figure supplement 1g*). Third, AS and DS bins showed comparable distributions of H4K20me3 levels at comparable RT and gene expression status (*Figure 6a, Figure 5—figure supplement 1g*). Interestingly, NRRs showed a specific, broader distribution of H4K20me3 levels, including a higher proportion of highly enriched windows, especially compared to URRs (compare boxplots in *Figure 6a*, *Figure 5—figure supplement 1g*). Locally, high densities of H4K20me3 are therefore detected not only in late, non-genic AS segments but also in late-replicating gene deserts of null RFD, which presumably replicate by widespread, spatially random initiation. This result led us to repeat the analysis of ORC/MCM enrichment at H4K20me3-high and -low 10 kb intergenic bins in NRRs. Again, ORC/MCM were more abundant at H4K20me3-high than -low bins (*Figure 6b*). These findings support the hypothesis that H4K20me3 facilitates origin licensing specifically in these heterochromatic segments (*Brustel et al., 2017*).

## Discussion

The study presented here provides a novel, comprehensive genome-wide analysis of multiple pre-RC proteins compared with RFD, transcription, and RT profiles in human cells. We find a widespread presence of ORC/MCM throughout the genome, with variations that only depend on RT or active transcription. ORC/MCM are depleted from transcribed genes and enriched at TSSs. ORC/MCM are more abundant in early than in late RTDs. The even distribution of ORC/MCM observed within IZs is consistent with OK-seq results, suggesting that initiation probability is fairly homogeneous within

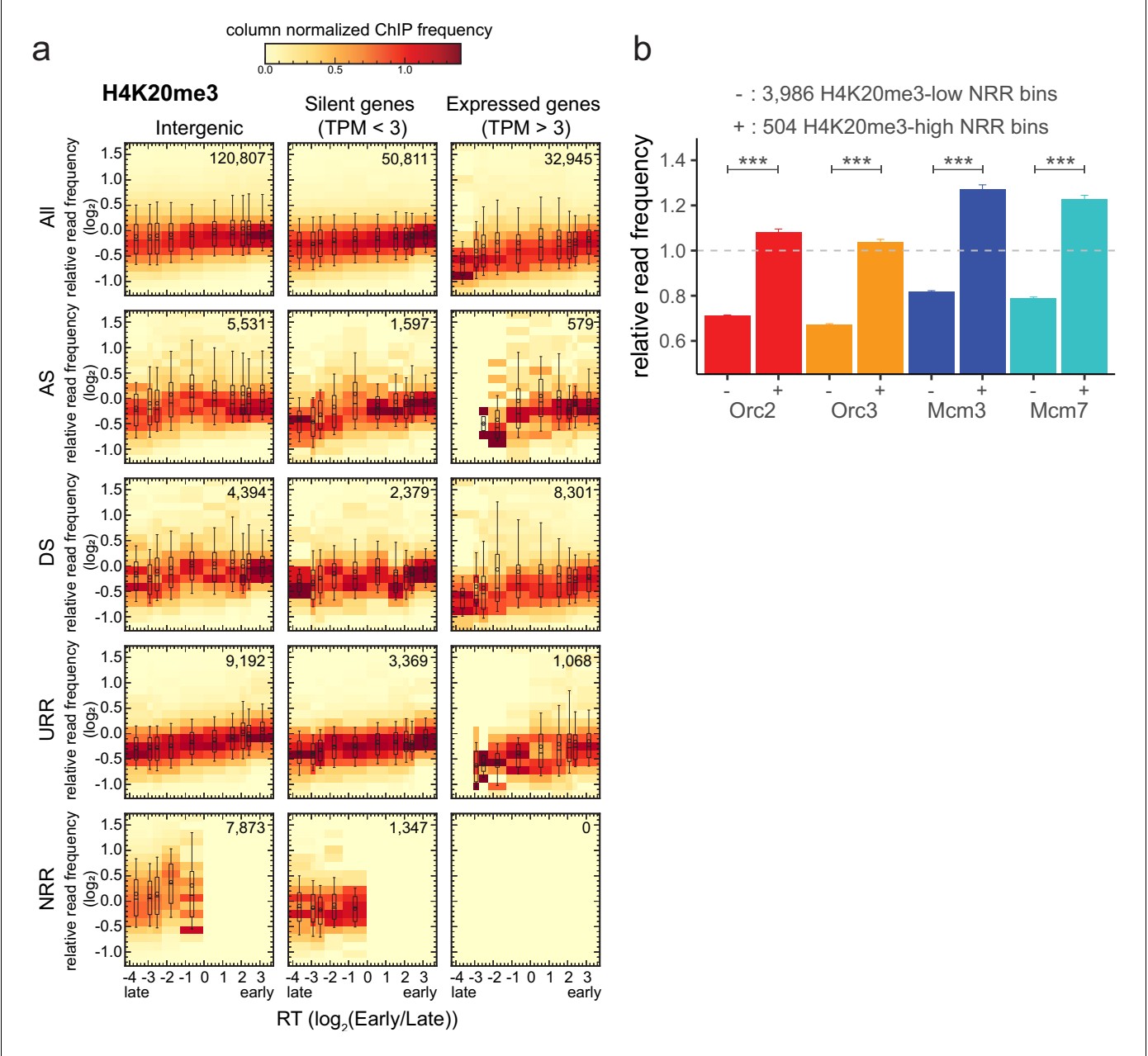

**Figure 6.** H4K20me3 is enriched in late-replicating null RFD regions (NRR). (**a**) 3 × 5 panel of 2D histograms of H4K20me3 chromatin immunoprecipitation relative read frequencies versus replication timing (RT) (average $\log_2$(Early/Late) over 100 kb binned according to the decile of RT distribution). The analysis was performed in 10 kb bins. Histograms are normalized by column and displayed for different bin categories (columns: intergenic regions, silent genes, expressed genes; rows: all bins, ascending segment [AS], descending segment [DS], unidirectionally replicating region [URR], null replication fork directionality region [NRR] bins) as for origin recognition complex/minichromosome maintenance complex (ORC/MCM) in **Figure 4**. The number of bins per histogram is indicated in each panel. Superimposed boxplots represent the mean (circle), median (thick line), first and third quartile (box), and first and ninth decile (whiskers) of the relative read frequencies in each timing bins. Refer to **Figure 5—figure supplement 1g** for statistical comparisons. (**b**) Histogram representation of mean ±2 × SEM of ORC/MCM relative read frequencies at 3986 H4K20me3-low NRR 10 kb bins and 504 H4K20me3-high NRR 10 kb bins. Statistics were performed using one-sided *t*-test. ***p<0.001.

IZs. However, when RT and transcriptional effects are controlled, no significant differences in ORC/MCM densities are detected between regions supporting either preferential replication initiation (ASs) or termination (DSs), or random replication (NRRs), or unidirectional, passive replication (URRs). We consequently propose that potential origins, defined by loaded MCM-DHs, are

widespread through the genome and that preferential initiation sites are selected for activation in S phase based on additional genetic and/or epigenetic factors. We further show that subsets of non-genic ASs and randomly replicating gene deserts are enriched in H4K20me3, which helps recruiting ORC/MCM to these late-replicating segments.

Our data suggest that transcription has both positive and negative effects on origin activity. Actively transcribed gene bodies are depleted of ORC/MCM (*Figure 2*, *Figure 3a*). As reported in *Drosophila* (*Powell et al., 2015*), we propose that active transcription removes ORC/MCM from transcribed gene bodies, which reduces their replication initiation capacity. This mechanism is consistent with previous studies reporting that replication does not initiate within transcribed genes (*Hamlin et al., 2010*; *Hyrien et al., 1995*; *Knott et al., 2009*; *Macheret and Halazonetis, 2018*; *Martin et al., 2011*; *Sasaki et al., 2006*).

By contrast, ORC and, to a lesser degree, MCM are enriched at active TSSs (*Figure 3*). Active TSSs are regions of open chromatin structure characterized by DNase- or MNase hypersensitivity. Such hypersensitivity is also a hallmark of Ma-IZs (*Boulos et al., 2015*; *Papior et al., 2012*) and preferred ORC binding sites (*Miotto et al., 2016*; *Petryk et al., 2016*). However, this increased ORC binding does not necessarily increase local initiation efficiency since the most efficient initiation sites identified by EdUseq-HU within early IZs are associated with poly(dA:dT) tracts, but not TSSs (*Tubbs et al., 2018*). Our finding that MCMs are less enriched at TSSs than ORC also argues against highly preferential origin licensing at TSSs. Furthermore, since MCMs are distributed fairly evenly upstream and downstream of transcribed gene bodies (*Figure 3a*), the preferred binding of ORC at TSSs does not result in increased MCM-DH loading specifically upstream of genes.

We previously reported that IZs are enriched in the histone variant H2A.Z and in open chromatin marks (H3K4me1, H3K27ac, DNAse HS sites) typical of active or poised enhancers (*Petryk et al., 2016*), which could potentially explain why IZs are more accessible to firing factors than flanking segments with comparable MCM-DH density. Recently, it was reported that H2A.Z recruits Suv420H1, which induces H4K20 dimethylation (*Long et al., 2020*). H4K20me2 interacts with the BAH domain of ORC1 (*Beck et al., 2012a*; *Kuo et al., 2012*; *Vermeulen et al., 2010*). The H2A.Z–H4K20me2–ORC1 axis therefore supports a role for H2A.Z in ORC recruitment and origin licensing (*Long et al., 2020*; *Petryk et al., 2016*). Furthermore, H3.3/H2A.Z double variant–containing nucleosomes present at active promoters and other regulatory regions constitute a less stable nucleosome that is more easily displaced, resulting in nucleosome-free gaps (*Jin et al., 2009*). Interestingly, nucleosome-free gaps associated with H3K4me2 are found at most binding sites for the origin-firing factor Treslin/MTBP, which may form looping interactions with distantly located MCM-DH to promote dispersed initiation within broad zones (*Kumagai and Dunphy, 2020*). These results provide novel insight into how multiple open chromatin marks previously detected within IZs may promote not only origin licensing, but also origin firing.

H4K20 methylation has multiple functions in ensuring genome integrity, such as DNA replication (*Beck et al., 2012b*; *Long et al., 2020*; *Picard et al., 2014*; *Tardat et al., 2010*), DNA repair, and chromatin compaction (*Jørgensen et al., 2013*; *Nakamura et al., 2019*; *Shoaib et al., 2018*), suggesting that the different functions are context-dependent and executed with different players. However, it is important to discriminate between H4K20me2, which is the most abundant H4K20 methylation state, and H4K20me3, which is more restricted (*Jørgensen et al., 2013*). We previously demonstrated that H4K20me3 provides a platform to enhance licensing in late-replicating heterochromatin (*Brustel et al., 2017*). We functionally link replication licensing to H4K20me3 in a specific subset of late-replicating domains as we detect both elevated ORC and MCM levels when selecting for H4K20me3-enriched non-genic ASs and NRRs (*Figures 5f* and *6b*). Whether H4K20me3 and/or additional chromatin modifications may also promote the origin-firing step remains to be investigated.

In higher eukaryotes, it has been proposed that RT could simply result from the spatial distribution of potential origins upon S phase entry. The latter distribution has been derived from ORC (*Dellino et al., 2013*; *Miotto et al., 2016*) or MCM-DH (*Das et al., 2015*; *Hyrien, 2016*) abundance, as well as from epigenetic mark profiles *Gindin et al., 2014*. For example, *Miotto et al., 2016* performed computational simulations where stochastic initiations at experimentally mapped ORC binding sites allow to reproduce human RT profiles. Our data also indicate a convincing correlation of ORC density with RT. However, we observed a weaker correlation of MCM-DH density with RT, and a lack of correlation with RFD slope, suggesting that origin-firing probability, and therefore RT, is

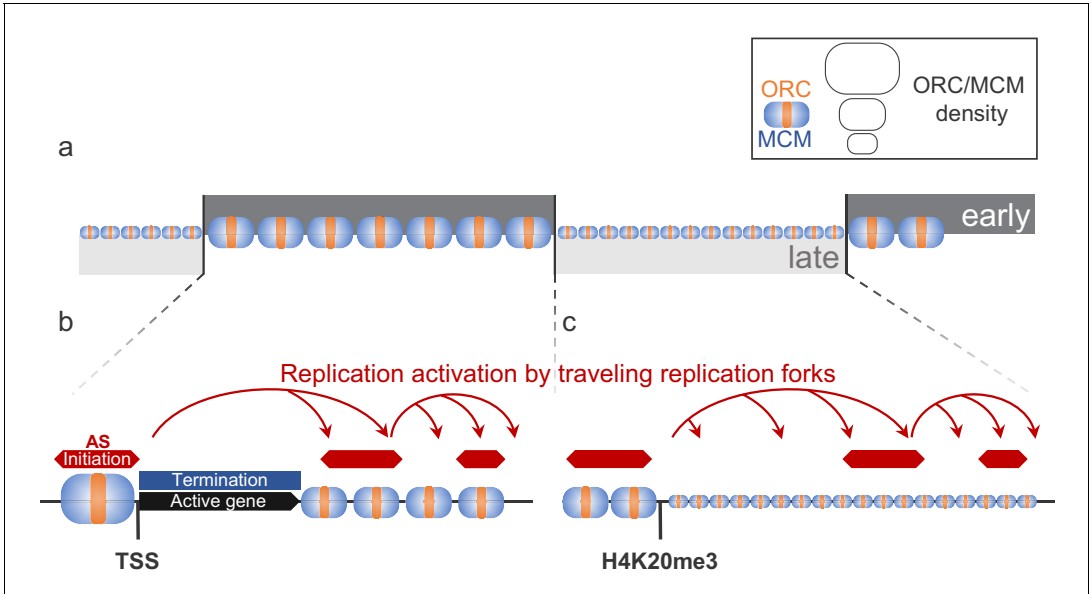

**Figure 7.** Model for replication organization in higher eukaryotes. (a) Replication is organized in large segments of constant replication timing (early replication timing domain [RTD], dark grey; late RTD, light grey) (*Marchal et al., 2019*). While we observe the ubiquitous presence of the origin recognition complex (ORC; orange) and the minichromosome maintenance complex (MCM; blue) throughout the genome, the enrichment levels of ORC/MCM were higher in early RTDs compared to late RTDs. (b) Early RTDs are among other characterized by active transcription. ORC/MCM are locally highly enriched at active transcription start site (TSS). However, actively transcribed gene bodies (black) are deprived of ORC/MCM, often correlating with replication termination (blue). Besides TSSs, we find ORC/MCM stochastically distributed along intergenic regions. We hypothesize that traveling replication forks trigger activation of replication in a cascade (red arrows). (c) In gene-deprived and transcriptionally silent late-replicating heterochromatin, we detected homogeneous ORC/MCM distribution at generally lower levels. H4K20me3 is present at late-replicating non-genic ascending segments (ASs) and null RFD regions (NRRs) and leads to enhanced ORC/MCM binding, linking this histone mark to replication activation in heterochromatin.

The online version of this article includes the following figure supplement(s) for figure 7:

**Figure supplement 1.** Early replication control elements (ERCEs) correlate with replication initiation.

not solely regulated by MCM-DH density (*Figure 4b*, *Figure 4—figure supplement 1e*, *Figure 5a*). The resolution of RT profiles is much less than RFD profiles, and it is not clear if models that predict RT would still correctly predict IZs. In fact, it was recently observed that all chromatin marks associated to open chromatin allowed very good predictions of RT profiles *Gindin et al., 2014*. Since open chromatin independently facilitates ORC binding in G1 phase and access of firing factors to MCM-DHs in S phase, open chromatin marks and ORC density may both predict RT without implying a direct causal link between RT and ORC binding. Probably, only the location of MCM-DHs associated with appropriate open chromatin marks to recruit firing factors is causative of RT.

The spatio-temporal replication program can change during cellular differentiation (*Marchal et al., 2019*). Comparison with chromatin conformation capture (Hi-C) data has shown that early and late RTDs correspond to the more and less accessible compartments of the genome, respectively (*Ryba et al., 2010*). Recently, Sima et al. used the CRISPR-Cas9 technology to identify three separate, cis-acting elements that together control the early replication time of the pluripotency-associated Dppa2/4 domain in mouse embryonic stem cells (mESCs) *Sima et al., 2019*. Strikingly, these early replication control elements (ERCEs) are enriched in CTCF-independent Hi-C interactions and active epigenetic marks (DNase1 HS, p300, H3K27ac, H3K4me1, H3K4me3) previously observed at OK-seq IZs (*Petryk et al., 2018*; *Petryk et al., 2016*). By mining mESC OK-seq data (*Petryk et al., 2018*), we found that the three ERCEs of the Dppa2/4 domain indeed fall within IZs (*Figure 7—figure supplement 1a*). Furthermore, the aggregate 1835 ERCEs predicted genome-wide by Sima et al., from epigenetic profiles of mESCs, shows a significant, positive RFD shift indicative of efficient replication initiation (*Figure 7—figure supplement 1b*). This finding is confirmed in proliferating PHA-stimulated primary splenic B cells (*Figure 7—figure supplement 1c*), attesting to the general validity of these observations. Since our data suggest that a higher ORC/MCM density *is*

*not* a distinguishing feature of IZs from the rest of the genome, IZ specification cannot solely occur at the origin-licensing step. Open and dynamic chromatin structures found at Ma-IZs and ERCEs (*Petryk et al., 2016*; *Sima et al., 2019*) might not only facilitate origin licensing in G1 but also promote chromatin binding of limiting firing factors during S phase (*Boos and Ferreira, 2019*; *Krude et al., 1997*; *Kumagai and Dunphy, 2020*).

## Conclusion

Our mapping of ORC and MCM complexes shows that in human cells most of the genome, except transcribed genes, is licensed for replication during the G1 phase of the cell cycle. ORC/MCM are more enriched in early than in late RTDs (*Figure 7a*) but only a fraction of MCM-DHs is selected for initiation during S phase. Open chromatin marks define efficient Ma-IZs, often but not always circumscribed by active genes (*Figure 7b*). Such marks may favor origin licensing in G1 but also binding of origin firing factors in S phase. In addition, H4K20me3 facilitates origin licensing in late-replicating regions (*Figure 7c*). Once forks emanate from Ma-IZs within an RTD, the omnipresence of MCM-DHs allows a cascade of replication initiation to take place dispersively between IZs (*Figure 7b*). The identification of ERCEs supports the hypothesis that open chromatin facilitates early origin activation. The links between chromatin structure and transcription, on the one hand, and origin licensing and activation, on the other hand, facilitate the timely activation of appropriate origins during programmed development.

## Materials and methods

### Key resources table

| Reagent type (species) or resource | Designation | Source or reference | Identifiers | Additional information |
|---|---|---|---|---|
| Cell line (*Homo sapiens*) | Raji (lymphoblast) | ATCC/DZMZ | ATCC: CCL-86 DZMZ: ACC 319 RRID:CVCL_0511 | B-lymphocyte; Burkitt's lymphoma Received from: https://www.dsmz.de/collection/catalogue/details/culture/ACC-319 Tested mycoplasma negative |
| Antibody | Cyclin A1/A2 (rabbit monoclonal) | Abcam | ab185619 | WB: 1:1000 |
| Antibody | Cyclin B1 (mouse monoclonal) | Abcam | ab72, RRID:AB_305751 | WB: 1:1000 |
| Antibody | H3S10P (rabbit monoclonal) | Cell Signaling | Clone D2C8, cat. no. 3377, RRID:AB_1549592 | WB: 1:1000 |
| Antibody | GAPDH (rat monoclonal) | This paper | Clone GAPDH3 10F4; Rat IgG2c | WB: 1:50 |
| Antibody | H4K20me1 (mouse monoclonal) | Diagenode | MAb-147-100 | ChIP: 2.5 µg |
| Antibody | H4K20me3 (rabbit polyclonal) | Diagenode | pAb-057-050, RRID:AB_2617145 | ChIP: 2.5 µg |
| Antibody | Rabbit-IgG (polyclonal) | Sigma | R2004, RRID:AB_261311 | ChIP: 10 µg |
| Antibody | Orc2 (rabbit polyclonal) | *Ritzi et al., 2003* | SA93 | Whole serum; ChIP: 15 µl |
| Antibody | Orc3 (rabbit polyclonal) | *Ritzi et al., 2003* | SA7976 | Whole serum; ChIP: 15 µl |
| Antibody | Mcm3 (rabbit polyclonal) | *Ritzi et al., 2003* | SA8413 | Whole serum; ChIP: 15 µl |
| Antibody | Mcm7 (rabbit polyclonal) | *Ritzi et al., 2003* | SA8496 | Whole serum; ChIP: 15 µl |

*Continued on next page*

*Continued*

| Reagent type (species) or resource | Designation | Source or reference | Identifiers | Additional information |
|---|---|---|---|---|
| Peptide, recombinant protein | Protein A Sepharose 4 Fast Flow | GE Healthcare | GE17-5280-11 | |
| Peptide, recombinant protein | Protein G Sepharose 4 Fast Flow | GE Healthcare | GE17-0618-06 | |
| Sequence-based reagent | *oriP* DS_fw | This paper | qPCR primers | 5′-AGTTCACTGCCCGCTCCT-3′ |
| Sequence-based reagent | *oriP* DS_rv | This paper | qPCR primers | 5′-CAGGATTCCACGAGGGTAGT-3′ |
| Sequence-based reagent | H4K20me1positive_fw | Eric Julien, personal communication | qPCR primers | 5′-ATGCCTTCTTGCCTCTTGTC-3′ |
| Sequence-based reagent | H4K20me1positive_rv | Eric Julien, personal communication | qPCR primers | 5′-AGTTAAAAGCAGCCCTGGTG-3′ |
| Sequence-based reagent | H4K20me3positive_fw | Eric Julien, personal communication | qPCR primers | 5′-TCTGAGCAGGGTTGCAAGTAC-3′ |
| Sequence-based reagent | H4K20me3positive_rv | Eric Julien, personal communication | qPCR primers | 5′-AAGGAAATGATGCCCAGCTG-3′ |
| Chemical compound, drug | Formaldehyde | Thermo Scientific | Prod# 28908 | 16% formaldehyde solution; methanol-free |
| Chemical compound, drug | Proteinase K | Roche | Cat. no. 03 115 852 001 | 1 mg/ml; 8 mg |
| Chemical compound, drug | RNase | Roche | Cat. no. 11 119 915 001 | 0.5 µg/µl; 2 µg |
| Commercial assay or kit | NucleoSpin Extract II Kit | Macherey-Nagel | Cat. no. 740609.50 | |
| Commercial assay or kit | Accel-NGS 1S Plus DNA Library Kit for Illumina | Swift Biosciences | Cat. no. 10096 | |
| Commercial assay or kit | Direct-zol™ RNA MiniPrep kit | Zymo Research | Cat. no. R2051 | |
| Commercial assay or kit | Encore Complete RNA-Seq Library Systems kit | NuGEN | Cat. no. 0333-32 | |
| Software, algorithm | Tophat2 | *Kim et al., 2013* | RRID:SCR_013035 | |
| Software, algorithm | HTSeq-count | *Anders et al., 2015* | RRID:SCR_011867 | |
| Software, algorithm | BWA (v.0.7.4) | *Li and Durbin, 2009* | RRID:SCR_010910 | OK-seq mapping, default parameters |
| Software, algorithm | bowtie (v.1.1.1) | *Langmead et al., 2009* | RRID:SCR_005476 | ChIP-seq mapping, bowtie -m one index *file*.fastq |
| Software, algorithm | deepTools (v.3.3.1) | *Ramírez et al., 2016* | RRID:SCR_016366 | |
| Software, algorithm | MACS2 (v.2.2.5) | *Zhang et al., 2008* | RRID:SCR_013291 | Default settings, `--broad` |
| Software, algorithm | R (v.3.2.3) | *R Development Core Team, 2018* | RRID:SCR_001905 | |
| Software, algorithm | dplyr (v.0.8.5) | *Wickham et al., 2020* | RRID:SCR_016708 | R package |
| Software, algorithm | ggplot2 (v.3.1.0) | *Wickham, 2016* | RRID:SCR_014601 | R package |
| Software, algorithm | gplots (v.3.0.3) | *Warnes et al., 2020* | | R package |
| Software, algorithm | Python 2.7 and Phyton 3 | *van Rossum, 1995* | RRID:SCR_008394 | |
| Software, algorithm | numpy (v.1.18.5) | *Harris et al., 2020* | RRID:SCR_008633 | Python library |
| Software, algorithm | matplotlib (v.3.2.3) | *Hunter, 2007* | RRID:SCR_008624 | Python library |
| Software, algorithm | SciPy (v.1.5.0) | *Virtanen et al., 2020* | RRID:SCR_008058 | Python library |

## Cell culture

Raji cells (ATCC: CCL-86; DZMZ: ACC 319) were directly obtained from DZMS and tested mycoplasma negative. Raji cells were cultured at 37°C and 5% $CO_2$ in RPMI 1640 (Gibco, Thermo Fisher, USA) supplemented with 8% FCS (Lot BS225160.5, Bio and SELL, Germany), 100 Units/ml penicillin, 100 µg/ml streptomycin (Gibco, Thermo Fisher, USA), 1× MEM non-essential amino acids (Gibco, Thermo Fisher, USA), 2 mM L-glutamine (Gibco, Thermo Fisher, USA), and 1 mM sodium pyruvate (Gibco, Thermo Fisher, USA).

## RNA extraction, sequencing, and TPM calculation

RNA-seq was performed in three independent replicates. RNA was extracted from $3 \times 10^5$ Raji cells using Direct-zol™ RNA MiniPrep kit (Zymo Research) according to manufacturer's instructions. RNA quality was confirmed by Bioanalyzer RNA integrity numbers between 9.8 and 10 followed by library preparation (Encore Complete RNA-Seq Library Systems kit [NuGEN]). Single-end 100 bp sequencing was performed by Illumina HiSeq 1500 to a sequencing depth of 25 million reads. The reads were mapped to hg19 genome using Tophat2 and assigned to annotated genes (HTSeq-count) (*Anders et al., 2015*; *Kim et al., 2013*). TPM values were calculated for each sample ($TPM_j = 10^6 \frac{n_j}{l_j} / \sum_i \frac{n_i}{l_i}$, where $n_i$ is the number of reads that map to gene $i$ whose total exon length expressed in kb is $l_i$) as previously described (*Wagner et al., 2012*).

## Replication fork directionality profiling using OK-seq method in Raji

Raji OK-seq was recently published and is available from the European Nucleotide Archive under accession number PRJEB25180 (see Data access section) (*Wu et al., 2018*). Reads > 10 nt were aligned to the human reference genome (hg19) using the BWA (version 0.7.4) software with default parameters (*Li and Durbin, 2009*). We considered uniquely mapped reads only and counted identical alignments (same site and strand) as 1 to remove PCR duplicate reads. Five replicates were sequenced providing a total number of 193.1 million filtered reads (between 19.1 and 114.1 million reads per replicate). RFD was computed as $RFD = \frac{(R-F)}{(R+F)}$, where 'R' (resp. 'F') is the number of reads mapped to the reverse (resp. forward) strand of the considered regions. RFD profiles from replicates were highly correlated, with Pearson correlation computed in 50 kb non-overlapping windows with >100 mapped reads (R + F) ranging from 0.962 to 0.993. Reads from the five replicate experiments were pooled together for further analyses.

## Determining regions of ascending, descending, and constant RFD

RFD profiling of two human cell lines revealed that replication primarily initiates stochastically within broad (up to 150 kb) zones and terminates dispersedly between them (*Petryk et al., 2016*). These IZs correspond to quasi-linear ASs) of varying size and slope within the RFD profiles. As previously described for mean RT profiles analysis (*Audit et al., 2013*; *Baker et al., 2012*), we determined the smoothed RFD profile convexity from the convolution with the second derivative of the Gaussian function of standard deviation 32 kb. In total, 4891 ASs were delineated as the regions between positive and negative convexity extrema of large amplitude. The amplitude threshold was set in a conservative manner in order to mainly detect the most prominent IZs as described and to avoid false positives *Petryk et al., 2016*. Descending segments (DSs) were detected symmetrically to ASs as regions between negative and positive convexity extrema using the same threshold. Noting *pos_5'* and *pos_3'* the location of the start and end position of an AS or DS segment, each segment was associated to its size *pos_3'-pos_5'* and the RFD shift across its length: ΔRFD = RFD (*pos_3'*) – RFD (*pos_5'*). DS segments were less numerous (2477 versus 4891) and on average larger (126 kb versus 38.8 kb) than AS segments, as expected, and presented a smaller average RFD shift (|ΔRFD| = 0.69 versus 0.83).

Initial RFD profiling in human also revealed regions of unidirectional fork progression and regions of null RFD where replication is bidirectional. URRs were delineated as regions where |ΔRFD| > 0.8 homogeneously over at least 300 kb (401 regions of mean length 442 kb covering 177 Mb). NRRs were delineated as regions where |ΔRFD| < 0.15 homogeneously over at least 500 kb (127 regions of mean length 862 kb covering 110 Mb). Thresholds were set in a conservative manner to avoid false positive, particularly not to confuse RFD zero-crossing segments with NRR.

## Centrifugal elutriation and flow cytometry

For centrifugal elutriation, $5 \times 10^9$ exponentially growing Raji cells were harvested, washed with PBS, and resuspended in 50 ml RPMI 1680, 8% FCS, 1 mM EDTA, 0.25 U/ml DNaseI (Roche, Germany). Concentrated cell suspension was passed through 40 µm cell strainer and injected in a Beckman JE-5.0 rotor with a large separation chamber turning at 1500 rpm and a flow rate of 30 ml/min controlled by a Cole-Parmer Masterflex pump. While rotor speed was kept constant, 400 ml fractions were collected at increasing flow rates (40, 45, 50, 60, and 80 ml/min). Individual fractions were quantified, $5 \times 10^6$ cells washed in PBS, ethanol fixed, RNase treated and stained with 0.5 mg/ml Propidium Iodide. DNA content was measured using the FL2 channel of FACSCalibur (BD Biosciences, Germany). Remaining cells were subjected to chromatin cross-linking.

## Generation of GAPDH monoclonal antibody

Rat monoclonal antibody GAPDH3 10F4 were generated by immunization with a peptide comprising amino acids RLEKPAKYDDIKKVVK of human GAPDH (aa246-263) coupled to OVA. Animals were injected subcutaneously and intraperitoneally with a mixture of 50 µg peptide, 5 nmol CpG (Tib Molbiol, Berlin, Germany), and an equal volume of incomplete Freund's adjuvant. Six weeks later a booster injection was performed without Freund's adjuvant. Three days later, spleen cells were fused with P3X63Ag8.653 myeloma cells using standard procedures. Hybridoma supernatants were screened in a solid-phase enzyme-linked immunosorbent assay for binding to GAPDH antigen. Positive supernatants were further assayed for western blotting. Hybridoma cells were subcloned twice by limiting dilution to obtain the monoclonal cell line stably producing antibody GAPDH3 10F4 (rat IgG2c).

## Chromatin cross-linking with formaldehyde

Raji cells were washed twice with PBS, resuspended in PBS to a concentration of $2 \times 10^7$ cells/ml, and passed through 100 µm cell strainer (Corning Inc, USA). Fixation for 5 min at room temperature was performed by adding an equal volume of PBS 2% methanol-free formaldehyde (Thermo Scientific, USA, final concentration: 1% formaldehyde) and stopped by the addition of glycine (125 mM final concentration). After washing once with PBS and once with PBS 0.5% NP-40, cells were resuspended in PBS containing 10% glycerol, pelleted, and snap frozen in liquid nitrogen.

## Cyclin western blot

Cross-linked samples were thawed on ice, then resuspended in LB3+ sonication buffer (see below) containing protease inhibitor and 10 mM MG132. After sonicating $3 \times 5$ min (30 s on, 30 s off) using Bioruptor in the presence of 212–300 µm glass beads, samples were treated with 50 U Benzonase for 15 min at room temperature and centrifuged 15 min at maximum speed. Also, 50 µg protein lysates were loaded on 10% SDS-polyacrylamide gel (cyclin A1/A2, cyclin B1), or 12.5–15% gradient gel (H3S10P). Cyclin A1/A2 (Abcam, ab185619), cyclin B1 (Abcam, ab72), and H3S10P (Cell Signaling, D2C8) antibodies were used in 1:1000 dilutions, and GAPDH (clone GAPDH3 10F4, rat IgG2c; this study) was diluted 1:50. HRP-coupled secondary antibodies were used in 1:10,000 dilutions. Detection was done using ECL on CEA Blue Sensitive X-ray films.

## Chromatin sonication

Cross-linked cell pellets were thawed on ice, then resuspended in LB3(+) buffer (25 mM HEPES [pH 7.5], 140 mM NaCl, 1 mM EDTA, 0.5 mM EGTA, 0.5% sarkosyl, 0.1% DOC, 0.5% Triton-X-100, 1× protease inhibitor complete [Roche, Germany]) to a final concentration of $2 \times 10^7$ cells/ml. Sonication was performed in AFA Fiber and Cap tubes ($12 \times 12$ mm, Covaris, Great Britain) at an average temperature of 5℃ at 100 W, 150 cycles/burst, 10% duty cycle, 20 min (S-G2-M fraction: 17 min) using the Covaris S220 (Covaris Inc, UK), resulting in DNA fragments of 100–300 bp on average.

## Chromatin immunoprecipitation and qPCR quality control

Sheared chromatin was pre-cleared with 50 µl protein A Sepharose 4 Fast Flow beads (GE Healthcare, Germany) per 500 µg chromatin for 2 hr. Then, 500 µg chromatin (or 250 µg for histone methylation) were incubated with rabbit anti-Orc2, anti-Orc3, anti-Mcm3, anti-Mcm7 (*Papior et al., 2012*), mouse anti-H4K20me1 (Diagenode, MAb-147-100), rabbit anti-H4K20me3 (Diagenode, pAb-057-

050), or IgG isotype controls for 16 hr at 4°C. BSA-blocked protein A beads (0.5 mg/ml BSA, 30 µg/ml salmon sperm, 1× protease inhibitor complete, 0.1% Triton-X-100 in LB3(-) buffer [without detergents]) were added (50 µl/500 µg chromatin) and incubated for at least 4 hr on an orbital shaker at 4°C. Sequential washing steps with RIPA-150mM NaCl (0.1% SDS, 0.5% DOC, 1% NP-40, 50 mM Tris [pH 8.0], 1 mM EDTA), RIPA-300 mM NaCl, RIPA-250 mM LiCl buffer, and twice in TE (pH 8.0) buffer were performed. Immunoprecipitated chromatin fragments were eluted from the beads by shaking twice at 1200 rpm for 10 min at 65°C in 100 µl TE 1% SDS. The elution was treated with 80 µg RNAse A for 2 hr at 37°C and with 8 µg proteinase K at 65°C for 16 hr. DNA was purified using the Nucleo-Spin Extract II Kit. Quantitative PCR analysis of the EBV *oriP* Dyad Symmetry element (for pre-RC ChIP) or H4K20me1 and -me3 positive loci were performed using the SYBR Green I Master Mix (Roche) and the Roche LightCycler 480 System. Oligo sequences for qPCR were DS_fw: AGTTCAC TGCCCGCTCCT, DS_rv: CAGGATTCCACGAGGGTAGT, H4K20me1positive_fw: ATGCCTTCTTGCC TCTTGTC, H4K20me1positive_rv: AGTTAAAAGCAGCCCTGGTG, H4K20me3positive_fw: TCTGAG-CAGGGTTGCAAGTAC, H4K20me3positive_rv: AAGGAAATGATGCCCAGCTG. Chromatin fragment sizes were verified by loading 1–2 µg chromatin on a 1.5% agarose gel. Samples were quantified using Qubit HS dsDNA assay.

## ChIP-sample sequencing

ChIP sample library preparations from >4 ng of ChIP-DNA was performed using Accel-NGS 1S Plus DNA Library Kit for Illumina (Swift Biosciences). A 50 bp single-end sequencing was done with the Illumina HiSEQ 1500 sequencer to a sequencing depth of ~70 million reads. Fastq-files were mapped against the human genome (hg19, GRCh37, version 2009), extended for the EBV genome (NC007605) using bowtie (v1.1.1) (*Langmead et al., 2009*). Sequencing profiles were generated using deepTools' bamCoverage function using reads extension to 200 bp and reads per genomic content normalization (*Ramírez et al., 2016*). Visualization was performed in UCSC Genome Browser (http://genome.ucsc.edu).

For H4K20me1 and -me3 ChIP-seq data, MACS2 peak-calling (*Zhang et al., 2008*) was performed using the broad setting and overlapping peaks in three replicates were retained for further analyses.

## Binning approach and normalization

All data processing and analysis steps were performed in R (v.3.2.3) and numpy (v.1.18.5) python library, and visualizations were done using the ggplot2 (v3.1.0) package (*R Development Core Team, 2018*) and matplotlib (v.3.2.3) python library. The numbers of reads were calculated in non-overlapping 1 or 10 kb bins and saved in bed files for further analysis. To combine replicates, their sum per bin was calculated (=read frequency). To adjust for sequencing depth, the mean frequency per bin was calculated for the whole sequenced genome and all bins' counts were divided by this mean value, resulting in the normalized read frequency. To account for variations in the input sample, we additionally removed bins without reads in the input from all samples and divided by the normalized read frequency of the input, resulting in the relative read frequency. When aggregating different loci, input normalization was performed after averaging. This resulted in relative read frequency ranging from 0 to ~30. Pairwise Pearson correlations of ORC/MCM samples were clustered by hierarchical clustering using complete linkage clustering.

## Relation of ChIP relative read frequencies to Orc2 (K562) and DNase hypersensitivity

Orc2 ChIP-seq data in asynchronously cycling K562 cells was retrieved from GSE70165 (*Miotto et al., 2016*). Peak calling using default MACS2 settings resulted in 16,767 detected peaks overlapping from two replicates.

The ENCODE 'DNase clusters' track wgEncodeRegDnaseClusteredV3.bed.gz (December 3, 2017) containing DNase hypersensitive sites from 125 cell lines were retrieved from *Thurman et al., 2012*. Bins overlapping or not with HS sites larger than 1 kb were defined and the respective ChIP read frequency assigned for comparison.

## Comparison of ChIP relative read frequencies to replication data

ASs were aligned on their left (5') and right (3') borders. Mean and standard error of the mean (SEM) of relative read frequencies of aligned 1 kb bins were then computed to assess the average ChIP signal around the considered AS borders 50 kb away from the AS to 10 kb within the AS as this was sufficient to visualize the full increase of ORC/MCM coverage when entering ASs – the ORC/MCM relative read frequency plateaus inside ASs in *Figure 2b-d* are clearly seen. To make sure bins within the ASs were closer to the considered AS border than to the opposite border, only ASs of size >20 kb were used (3247/4891). We also limited this analysis to ASs corresponding to efficient IZs by requiring ΔRFD > 0.5, filtering out a further 290 lowly efficient ASs, leaving 2957 ASs for the analyses (*Table 1*).

In order to interrogate the relationship between ASs and transcription, we compared the results obtained for different AS groups: 506 ASs were classified as non-genic AS when the AS locus extended 20 kb at both ends did not overlap any annotated gene; the remaining 2451 ASs were classified as genic ASs. From the latter group, 673 ASs were classified as type 1 ASs when both AS borders were flanked by at least one actively transcribed gene (distance of both AS borders to the closest transcribed [TPM >3] gene body was <20 kb), and 1026 ASs were classified as type 2 ASs when only one AS border was associated to a transcribed gene (*Table 1*).

In order to assess the role of H4H20me3 mark on AS specification, we also classified non-genic ASs depending on their input-normed H4K20me3 relative read frequency. We grouped the non-genic ASs where the H4K20me3 relative read frequency was above the genome mean value by more than 1.5 standard deviation (estimated over the whole genome) and the non-genic ASs where the H4K20me3 relative read frequency was below the genome mean value. This resulted in 154 non-genic ASs with H4K20me3 signal significantly higher than genome average and 242 non-genic ASs with H4K20me3 signal lower than genome average.

A similar selection was performed on fully intergenic 10 kb windows within NRRs (as done above using the mean and standard deviation of H4K20me3 relative read frequency estimated on all fully intergenic 10 kb windows). This resulted in 504 and 3986 windows with high and low H4K20me3 signal, respectively.

## Comparison of ChIP relative read frequencies to transcription data

Gene-containing bins were determined and overlapping genes removed from the analysis. For cumulative analysis, we only worked with genes larger 30 kb and assigned the gene expression levels in TPM accordingly. Genes were either aligned at their TSS or their TTS, and the corresponding average ChIP read frequency windows were calculated in a 30 kb window centered on the alignment site.

## Comparison of ChIP relative read frequencies to RT

For identification of RTDs in Raji cells, we used the early- to late-RT ratio determined by Repli-seq (*Sima et al., 2018*). We directly worked from the pre-computed early to late log ratio from supplementary file GSE102522_Raji_log2_hg19.txt downloaded from GEO (accession number GSE102522). The timing of every non-overlapping 10 kb bin was calculated as the averaged $\log_2$(Early/Late) ratio within the surrounding 100 kb window. Early RTDs were defined as regions where the average log ratio >1.6 and late RTDs as regions where the average log ratio <−2.0. These thresholds resulted in 1648 early RTDs, ranging from 10 to 8940 kb in size, with a mean size of 591 kb, while we detected 2046 late RTDs in sizes from 10 to 8860 kb, averaging at 470 kb. These RTDs were used to classify ChIP read relative frequencies calculated in 10 kb bins as early or late RT. Bins overlapping any gene extended by 10 kb on both sides were removed from the analysis to avoid effects of gene activity on ChIP signals.

## Comparison of ChIP relative read frequencies distributions at different RT depending on transcriptional and replicative status

All non-overlapping 10 kb windows were classified as intergenic if closest genes were more than 5 kb away, as belonging to a silent (resp. expressed) gene body if the window was inside a gene with TPM <3 (resp. TPM >3) and at more than 3 kb of gene borders, otherwise windows were disregarded. This made sure that specific ChIP signal at gene TSS and TTS was not considered in the

analysis. Using the three window categories, we computed the 2D histograms of ChIP relative read frequencies versus RT in intergenic, silent and expressed gene bodies. We used 10 timing bins corresponding to the deciles of the whole genome timing distribution. For each timing bin, the histogram counts were normalized so as to obtain an estimate of the probability distribution function of the ChIP signal at the considered RT. The analysis was reproduced after restricting for windows fully in (i) AS segments (size >20 kb, $\Delta$RFD > 0.5), (ii) DS segments (size >20 kb, $\Delta$RFD < $-0.5$), (iii) URRs, and (iv) NRRs.

## Statistics

Statistical analyses were performed in R using one-sided $t$-test with Welch correction and 95% confidence interval or one-way ANOVA followed by Tukey's multiple comparisons of means with 95% family-wise confidence level, if appropriate. Comparison between ChIP signal distribution observed in two situations was performed computing the two-sample Kolmogorov–Smirnov statistics $D_{KS}$ using SciPy (v.1.5.0) statistical library and correcting for sample sizes by reporting $Z_{KS} = D_{KS}\sqrt{\frac{nm}{n+m}}$, where $n$ and $m$ are the sizes of the two samples, respectively.

## ERCE RFD profiles

The positions of the three genetically identified ERCEs in the mESC Dppa2/4 locus and of the 1835 predicted mESC ERCEs were downloaded from *Sima et al., 2019*. The mESC OK-seq data were downloaded from *Petryk et al., 2018* (SRR7535256) and mapped to mm10 genome (*Petryk et al., 2018*). OK-seq data from cycling mouse B cells were downloaded from *Tubbs et al., 2018* (GSE116319). The RFD profile was computed as in *Hennion et al., 2020* with 10 kb binning steps. Predicted ERCE shuffling was performed using a homemade function keeping the number of ERCE constant for each chromosome and avoiding unmapped genome sequences (genome regions with >20 consecutive Ns). Aggregated average RFD profiles were centered on the ERCE. The profile's envelopes represent the 95% confidence interval based on the mean and standard deviation at each position.

## Acknowledgements

We thank Tobias Straub for initial help with bioinformatical analyses, Torsten Krude for critical comments on the manuscript, and Hadi Kabalane for help with Raji RFD data.

AS was supported by the Deutsche Forschungsgemeinschaft (SFB 1064 TP05), SPP1230, and the HELENA graduate school of the Helmholtz Zentrum München. BA and OH were supported by the Agence Nationale de la Recherche (ANR-15-CE12-0011, ANR-18-CE45-0002, ANR-19-CE12-0028) and the Fondation pour la Recherche Médicale (FRM DEI201512344404), and the Cancéropôle Ile-de-France and the INCa (PL-BIO16-302). OH was supported by the Ligue Nationale Contre le Cancer (Comité de Paris; RS19/75-75), the Association pour la Recherche sur le Cancer (PJA 20171206387), and the program 'Investissements d'Avenir' launched by the French Government and implemented by the ANR (ANR-10-IDEX-0001-02 PSL*Research University). WH was supported by the Deutsche Forschungsgemeinschaft (SFB1064/TP A13, SFB-TR36/TP A04), Deutsche Krebshilfe (grant number 70112875), and National Cancer Institute (grant number CA70723). Publication costs were covered by the SFB 1064.

## Additional information

### Funding

| Funder | Grant reference number | Author |
| --- | --- | --- |
| Helmholtz Zentrum München (GmbH) | | Nina Kirstein<br>Alexander Buschle<br>Wolfgang Hammerschmidt<br>Aloys Schepers |
| Deutsche Forschungsgemeinschaft | SFB 1064/TP05 | Aloys Schepers |
| Agence Nationale de la Re- | ANR-15-CE12-0011 | Olivier Hyrien |

| | | |
|---|---|---|
| cherche | | Benjamin Audit |
| Fondation pour la Recherche Médicale | FRM DEI201512344404 | Olivier Hyrien Benjamin Audit |
| Cancéropôle Île-de-France | PL-BIO16-302 | Olivier Hyrien Benjamin Audit |
| Ligue Contre le Cancer | RS19/75-75 | Olivier Hyrien |
| Association pour la Recherche sur le Cancer | PJA 20171206387 | Olivier Hyrien |
| Deutsche Krebshilfe | 70112875 | Wolfgang Hammerschmidt |
| National Cancer Institute | CA70723 | Wolfgang Hammerschmidt |
| Agence Nationale de la Recherche | ANR-18-CE45-0002 | Olivier Hyrien Benjamin Audit |
| Agence Nationale de la Recherche | ANR-19-CE12-0028 | Olivier Hyrien Benjamin Audit |
| Agence Nationale de la Recherche | ANR-10-IDEX-0001-02 | Olivier Hyrien |
| Deutsche Forschungsgemeinschaft | SFB1064/TP A13 | Wolfgang Hammerschmidt |
| Deutsche Forschungsgemeinschaft | SFB-TR36/TP A04 | Wolfgang Hammerschmidt |
| Deutsche Forschungsgemeinschaft | SPP1230 | Aloys Schepers |

The funders had no role in study design, data collection and interpretation, or the decision to submit the work for publication.

## Author contributions

Nina Kirstein, Conceptualization, Data curation, Formal analysis, Investigation, Visualization, Methodology, Writing - original draft, Writing - review and editing; Alexander Buschle, Data curation, Formal analysis, Investigation; Xia Wu, Data curation, Investigation; Stefan Krebs, Data curation; Helmut Blum, Resources, Supervision; Elisabeth Kremmer, Resources; Ina M Vorberg, antibody generation and alidation; Wolfgang Hammerschmidt, Conceptualization, Supervision, Funding acquisition; Laurent Lacroix, Formal analysis, Visualization; Olivier Hyrien, Conceptualization, Supervision, Funding acquisition, Validation, Writing - original draft, Writing - review and editing; Benjamin Audit, Conceptualization, Software, Formal analysis, Supervision, Funding acquisition, Validation, Investigation, Writing - original draft, Project administration, Writing - review and editing; Aloys Schepers, Conceptualization, Formal analysis, Supervision, Writing - original draft, Project administration, Writing - review and editing

## Author ORCIDs

Nina Kirstein ⓘ https://orcid.org/0000-0001-6030-6173
Ina M Vorberg ⓘ http://orcid.org/0000-0003-0583-4015
Wolfgang Hammerschmidt ⓘ http://orcid.org/0000-0002-4659-0427
Olivier Hyrien ⓘ https://orcid.org/0000-0001-8879-675X
Benjamin Audit ⓘ https://orcid.org/0000-0003-2683-9990
Aloys Schepers ⓘ https://orcid.org/0000-0002-5442-5608

## Decision letter and Author response

Decision letter https://doi.org/10.7554/eLife.62161.sa1
Author response https://doi.org/10.7554/eLife.62161.sa2

## Additional files

### Supplementary files

• Transparent reporting form

### Data availability

Sequencing data have been deposited in the European Nucleotide Archive (ENA) and NCBI Gene Expression Omnibus as indicated — ChIP-Seq: PRJEB32855, RNA-seq Raji: PRJEB31867, OK-seq Raji: PRJEB25180, Repli-seq Raji: GSE102522, OK-seq mESC: SRR7535256, OK-seq mouse B-cells: GSE116319. All data generated or analysed during this study are included in the manuscript and supporting files. Source data files have been provided for Figures 1–6, Figure 1–Supplements 1–3, Figure 2–Supplements 1,2; Figure 3–Supplement 1; Figure 4–Supplements 1,2; Figure 5–Supplement 1, and Figure 7–Supplement 1.

The following dataset was generated:

| Author(s) | Year | Dataset title | Dataset URL | Database and Identifier |
|---|---|---|---|---|
| Kirstein N, Buschle A, Wu X, Krebs S, Blum H, Hammerschmidt W, Lacroix L, Hyrien O, Audit B, Schepers A | 2020 | Human ORC/MCM density is low in active genes and correlates with replication time but does not delimit initiation zones | https://www.ebi.ac.uk/ena/browser/view/PRJEB32855 | European Nucleotide Archive (ENA), PRJEB32855 |

The following previously published datasets were used:

| Author(s) | Year | Dataset title | Dataset URL | Database and Identifier |
|---|---|---|---|---|
| Buschle A, Mrozek-Gorska P, Krebs S, Blum H, Cernilogar FM, Schotta G, Pich D, Straub T, Hammerschmidt W | 2019 | RNA-seq in Raji cells with inducible BZLF1 prior to and after induction of EBV's lytic cycle by doxycycline | https://www.ebi.ac.uk/ena/browser/view/PRJEB31867 | European Nucleotide Archive (ENA), PRJEB31867 |
| Wu X, Kabalane H, Kahli M, Petryk N, Laperrousaz B, Jaszczyszyn Y, Drillon G, Nicolini FE, Perot G, Robert A, Fund C, Chibon F, Xia R, Wiels J, Argoul F, Maguer-Satta V, Arneodo A, Audit B, Hyrien O | 2018 | Developmental and cancer-associated plasticity of DNA replication preferentially targets GC-poor, lowly expressed and late-replicating regions | https://www.ebi.ac.uk/ena/browser/view/PRJEB25180 | European Nucleotide Archive (ENA), PRJEB25180 |
| Sima J, Bartlett DA, Gordon MR, Gilbert DM | 2018 | Bacterial artificial chromosomes establish replication timing and sub-nuclear compartment de novo as extra-chromosomal vectors [repli-seq] | https://www.ncbi.nlm.nih.gov/geo/query/acc.cgi?acc=GSE102522 | NCBI Gene Expression Omnibus, GSE102522 |
| Petryk N, Dalby M, Wenger A, Stromme CB, Strandsby A, Andersson R, Groth A | 2018 | MCM2 promotes symmetric inheritance of modified histones during DNA replication | https://www.ebi.ac.uk/ena/browser/view/SRR7535256 | European Nucleotide Archive (ENA), SRR7535256 |
| Tubbs A, Sridharan S, van Wietmarschen N, Maman Y, Callen E, Stanlie A, Wu W, Wu X, Day A, Wong N, Yin M, | 2018 | OK-seq profile from cycling (S) phase untreated B cells | https://www.ncbi.nlm.nih.gov/geo/query/acc.cgi?acc=GSE116319 | NCBI Gene Expression Omnibus, GSE116319 |

Canela A,  Fu H,
Redon C,  Pruitt SC,
Jaszczyszyn Y,
Aladjem MI,  Aplan
PD,  Hyrien O,
Nussenzweig A

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
