## [Decision Letter]

**Acceptance summary:**

The manuscript characterizes the chromatin binding of two ORC subunits and two subunits of the MCM2-7 hexamer that are required for the initiation of DNA replication in different domains of the genome that are defined by the timing of DNA replication during S phase, early versus late. The binding of ORC and MCM subunits was compared with replication fork direction, transcription and replication timing profiles in human cells and subtle changes in the densities of these proteins in the different chromatin regions were observed. The distribution of these subunits, however, does not determine replication timing. The distribution in the genome of the histone H4K20me3 modification was also examined, indicating that it facilitates origin licensing in late-replicating regions. The authors suggest that factors other than the density and distribution of pre-Replicative Complexes determine the timing of the initiation of DNA replication during S phase.

**Decision letter after peer review:**

Thank you for submitting your article "Human ORC/MCM density is low in active genes and correlates with replication time but does not delimit initiation zones" for consideration by *eLife*. Your article has been reviewed by three peer reviewers, including Bruce Stillman as the Reviewing Editor and Reviewer #1, and the evaluation has been overseen by Kevin Struhl as the Senior Editor.

The reviewers have discussed the reviews with one another and the Reviewing Editor has drafted this decision to help you prepare a revised submission.

As the editors have judged that your manuscript is of interest, but as described below that additional experiments and analysis are required before it will be re-considered for publication.

Summary:

The authors have performed an extensive analysis of the binding or ORC2 and ORC3 subunits of the Origin Recognition Complex (ORC) and the MCM3 and MCM7 subunits of the MCM2-7 helicase subunits by chromatin immunoprecipitation and then compared the data to replicating timing patterns throughout the genome of human Raji cells. Based on previous studies, some of which derive from the authors' labs, they have identified different regions of the genome that replicate at different times and these are associated or not with transcription start sites (TSS; early replicating), with non-transcribed DNA and regions that replicate unidirectionally or have no preference in a population of cells. Now the authors correlate the MCM2-7 and ORC binding with the replication timing and the various categories of chromatin.

They conclude that ORC and MCM binding correlates with each other in G1 phase, which is not surprising, and then that early replicating regions are more associated with domains enriched in ORC/MCM binding, and uniquely, that there are large regions of homogeneous distributions of ORC/MCM. They conclude that ORC/MCM correlates with replication timing but not the probability of replication initiation. They also demonstrate that histone H4K20me3 location is localized with ORC/MCM and that a certain late replicating DNA has higher H4K20me3 binding.

The paper contains a great deal of work and is of interest to those in the DNA replication field, adding to what is already known. There are some papers that are either not discussed well or not even mentioned and this should be corrected. However, before the paper can be re-considered for publication, the authors need to address some major concerns:

Essential revisions

1) One major issue relates to the small enrichments of ORC/MCM they observe in early versus late replicating DNA and the possibility that this represents a DNA isolation bias rather than a real correlation or cause (see specific points 4, 5 and 6 below). The paper does not address this issue which, given the small differences in ORC/MCM between early and late replicating regions (1.4 fold), this needs to be addressed. It is understood that genome-wide mapping of pre-RC components in mammalian cells is challenging. In all of the studies to date, the ChIP enrichment is very modest and not confined to tight peaks that are typical of transcription factor binding. The weak and broad patterns of localization and their enrichment at hypersensitive and transcription start sites may be a technical artifact or is reflective of the underlying biology. While the signals they are observing are likely biological, it is still very difficult (as the authors allude to in the Discussion) to disentangle causation and correlation with the observed patterns and enrichment at transcription start sites and DNase HS sites. What would make this story much stronger is to demonstrate that the MCM2-7 signal is dynamic – that is that the enrichment patterns they observe in G1 should be very different from the patterns in late S-phase or G2 when replication forks have displaced most of the MCMs. The authors need to perform ChIP-seq on the cells elutriated at 80 ml/min. The ORC profile may also change due to the dynamic nature of ORC in human cells, but the MCM definitely should only be enriched in late replicating regions of the genome in late S phase and this comparison is needed.

2) As shown in Figure 1A, stochastic variation can be observed for MCM3/7 and ORC2/3 ChIP-seq replicates, it's reasonable to speculate that the input signal can also fluctuate randomly among replicates. Moreover, most of the conclusions in this manuscript are based on the input normalized signals. However, as shown by Figure 1A and the record for ENA PRJEB32855, no replication is performed for the input. Thus, we suggest that the authors provide replicates for the input, and normalize the ChIP signal to pooled input signals.

3) As shown in Figure 1—figure supplement 2A, the input signals at "DNase hypersensitive (HS)" is lower than those at "no HS"; and in Figure 1—figure supplement 3E, the ChIP signals of MCM3/7 and ORC2/3 at "HS" are higher than those in "no HS". Thus, when the ChIP signals of MCM3/7 and ORC2/3 are normalized to the input, will the difference of ChIP signals of McCM3/7 or ORC2/3 at "HS" and "no HS" be amplified artificially?

Additional comments related to the above major comments described above:

1) In Figure 1, the authors show a convincing correlation between the ChIP-seq profiles of ORC2 and ORC3, as well as between MCM3 and MCM7. Miotto et al. had published ORC2 ChIP-seq data using asynchronous K562 human erythroid cells. The data in the Kirstein et al. paper reports ChIP data of ORC2 form Raji lymphoblastoid cells. Although the cell types and cell cycle stage are different, it would be valuable to show a Pearson correlation between the two different ORC2 sets. This should be shown as a Supplement to Figure 1. The authors could also comment on the data of ORC1 ChIP from Dellino et al., 2013 and Long et al., 2019 (see point 8 below) whether these ORC patterns correlate with the other ORC ChIP data.

2) Figure 5A and Figure 4—figure supplement 2B. The results show that ORC is 1.4 times more frequently found in early versus late replicating regions. It is possible that the chromatin in early replicating regions is more accessible to the ChIP procedure than late replicating regions, which are likely more compact and hence difficult to access using antibodies. How have the authors excluded the possibility that extraction of DNA fragments in early versus late replicating regions could explain the difference in ORC binding? It should be noted that the authors previous papers and the Discussion in this paper claims accessibility to chromatin by replication factors may explain replication timing, yet they have assumed that the sonication and antibodies used for ChIP analysis are equally accessible. It is known that heterochromatic regions of the genome form phase transitions that may behave completely differently than actively transcribed and "accessible" regions of the genome.

3) Figure 5E. The same concern outlined in comment 2 above could explain the small, albeit statistically significant difference between H4K20me3 high and low regions of the genome.

4) Figure 6A and related text. The very slight differences in H4K20me3 levels could also be explained by extraction artIfact.

5) “However, potential origins are defined by assembled MCM-DHs, not by ORC”. This statement that potential origins are determined by the MCM2-7 DH and not by ORC is not logical because MCM2-7 DH is loaded by ORC and other factors. The idea that it correlates with ORC is dismissed a few lines later, but none of these statements are justified. What is the evidence that access of firing factors to MCM2-7 DH are regulated by chromatin access?

6) In a significant paper describing ORC ChIP and replication initiation, it was shown that ORC binding correlates with histone H2AZ and this could explain early replication origin activity. This paper is not even cited, much less discussed, and it should be (see Long et al., 2019.

7) The authors have dismissed the replication timing model proposed by Miottto et al., 2016, but it is not clear why. This model should be discussed in relationship to the model in Figure 7.

[Editors' note: further revisions were suggested prior to acceptance, as described below.]

Thank you for submitting your article "Human ORC/MCM density is low in active genes and correlates with replication time but does not delimit initiation zones" for consideration by *eLife*. Your article has been reviewed by three peer reviewers, including Bruce Stillman as the Reviewing Editor and Reviewer #1, and the evaluation has been overseen by Kevin Struhl as the Senior Editor.

The reviewers have discussed your response to the reviews with one another and the Reviewing Editor has drafted this decision to help you prepare a revised submission that addresses one issue.

The revised paper has incorporated new data that compares the abundance of ORC2, ORC3, MCM3 and MCM7 protein at two difference stages of the cell division cycle and raises some interesting observations. The authors have extensively addressed all of the original reviewer comments and provide new analysis. The differences in MCM and ORC levels at the different classes of gene expression, except for the gene bodies, are very modest but nonetheless statistically significant.

The general conclusion is that ORC is wide spread on the genome in G1 phase and MCM localizes with sites of initiation of DNA replication, and that histone H3K4me3 is correlated with late origin firing. At all locations, the exact site of initiation of DNA replication is stochastic.

One paradox needs explaining that arises from the new data presented in the revised paper compared to data in the literature.

1) The data in Figure 4—figure supplement 2C show that MCM3 and MCM7 levels are reduced in S-G2-M cells compared to G1 cells, but there remains a difference between early versus late replication timing domains in both cell cycle stages. In contrast, ORC is high in early RFDs and low in late RFDs at both stages. Perhaps the authors should discuss the significance of this result, in light of the fact that ORC1 is degraded in human cells at the G1-S transition and should not be present later in the cell cycle util it is re-synthesized. Does this mean ORC2 and ORC3 remain chromatin bound during the cell cycle and what does this mean.

We suggest that the authors address this issue in the Discussion of a revised manuscript which can then proceed.

---

## [Author Response]

Essential revisions1) One major issue relates to the small enrichments of ORC/MCM they observe in early versus late replicating DNA and the possibility that this represents a DNA isolation bias rather than a real correlation or cause (see specific points 4, 5 and 6 below). The paper does not address this issue which, given the small differences in ORC/MCM between early and late replicating regions (1.4 fold), this needs to be addressed. It is understood that genome-wide mapping of pre-RC components in mammalian cells is challenging. In all of the studies to date, the ChIP enrichment is very modest and not confined to tight peaks that are typical of transcription factor binding. The weak and broad patterns of localization and their enrichment at hypersensitive and transcription start sites may be a technical artifact or is reflective of the underlying biology. While the signals they are observing are likely biological, it is still very difficult (as the authors allude to in the Discussion) to disentangle causation and correlation with the observed patterns and enrichment at transcription start sites and DNase HS sites. What would make this story much stronger is to demonstrate that the MCM2-7 signal is dynamic – that is that the enrichment patterns they observe in G1 should be very different from the patterns in late S-phase or G2 when replication forks have displaced most of the MCMs. The authors need to perform ChIP-seq on the cells elutriated at 80 ml/min. The ORC profile may also change due to the dynamic nature of ORC in human cells, but the MCM definitely should only be enriched in late replicating regions of the genome in late S phase and this comparison is needed.

We thank the reviewers for this well-argued comment. We performed ORC and MCM ChIP-seq from elutriation fraction 80 ml/min, hereafter referred to as late S-G2-M (as determined in Figure 1—figure supplement 1) and analyzed their distribution with respect to replication timing.

In G1-derived chromatin, we previously observed both higher ORC and MCM binding in early RTDs compared to late RTDs (Figure 4, Figure 4—figure supplement 1D, E, Figure 5A and B, Figure 4—figure supplement 2B). In S-G2-M samples, ORC was still enriched in early compared to late RTDs. However, MCM signals were reduced in early compared to late RTDs, especially Mcm3. The early/late and G1/S-G2-M ratios ORC/MCM ChIP relative read frequencies are presented in Table 2. Early RTDs bind more ORC and MCM than late RTDs in G1, but this tendency is reduced or abolished in S-G2-M. Statistical testing validates that both ORC and MCM signals are dynamic between G1 and S-G2-M as predicted by the reviewers.

The cell-cycle dependent dynamic binding of ORC and MCM clearly suggests that the ORC/MCM enrichments observed in G1 in early compared late RTDs are of biological nature and not a technical artifact. We included the S-G2-M chromatin analyses in updated Figure 5B and the data without input normalization in updated Figure 4—figure supplement 2C.

2) As shown in Figure 1A, stochastic variation can be observed for MCM3/7 and ORC2/3 ChIP-seq replicates, it's reasonable to speculate that the input signal can also fluctuate randomly among replicates. Moreover, most of the conclusions in this manuscript are based on the input normalized signals. However, as shown by Figure 1A and the record for ENA PRJEB32855, no replication is performed for the input. Thus, we suggest that the authors provide replicates for the input, and normalize the ChIP signal to pooled input signals.

The reviewers are correct, we indeed did not show the input in replicates, as we were working with pooled input to obtain a higher read coverage. However, we performed input sequencing in three replicates and now show input replicates in Figure 1A. We uploaded the according input files to ENA PRJEB32855.

3) As shown in Figure 1—figure supplement 2A, the input signals at "DNase hypersensitive (HS)" is lower than those at "no HS"; and in Figure 1—figure supplement 3E, the ChIP signals of MCM3/7 and ORC2/3 at "HS" are higher than those in "no HS". Thus, when the ChIP signals of MCM3/7 and ORC2/3 are normalized to the input, will the difference of ChIP signals of McCM3/7 or ORC2/3 at "HS" and "no HS" be amplified artificially?

We indeed observed differential normalized read frequencies for the input when comparing “HS” to “no HS” (manuscript Figure 1—figure supplement 2B, Figure 1—figure supplement 3G).

When analyzing ORC and MCM normalized read frequencies without input division, we still observe a significant enrichment of ORC and MCM at HS sites. However, the enrichment is clearly more prominent for ORC, which is in line with previously reported Orc2 ChIP (Miotto et al., 2016). We do not necessarily expect the same observation for MCM due to their more disperse distribution.

We added DNase HS data without input normalization as new Figure 1—figure supplement 3G to the manuscript.

Input division is a widely used approach for ChIP-seq data analysis, because it also corrects for chromatin solubilization efficiencies between experiments. However, in our manuscript, we were aware of a potential bias introduced by the input. This is of relevance if the signal heights are low, as is the case for ORC and MCM. To visualize the impact of input normalization on the ORC/MCM profiles, we additionally included all analyses without input normalization in the figure supplements. When comparing both analysis approaches, our conclusions do not change qualitatively, although we do observe minor quantitative differences. We explain these important controls in the Results and mention figures comparing data with or without input normalization throughout the text.

Additional comments related to the above major comments described above:1) In Figure 1, the authors show a convincing correlation between the ChIP-seq profiles of ORC2 and ORC3, as well as between MCM3 and MCM7. Miotto et al. had published ORC2 ChIP-seq data using asynchronous K562 human erythroid cells. The data in the Kirstein et al. paper reports ChIP data of ORC2 form Raji lymphoblastoid cells. Although the cell types and cell cycle stage are different, it would be valuable to show a Pearson correlation between the two different ORC2 sets. This should be shown as a Supplement to Figure 1. The authors could also comment on the data of ORC1 ChIP from Dellino et al., 2013 and Long et al., 2019 (see point 8 below) whether these ORC patterns correlate with the other ORC ChIP data.

Our ORC/MCM ChIP-seq was established using optimized cross-linking time and mild sonication settings to avoid disruption of complexes on chromatin. The Covaris ultrasonicator S220 focuses the energy on the sample, allowing a considerably reduced applied power. These carefully established settings and the use of different antibodies to IP ORC- and MCM-proteins were presumably also the reason why we observed the dispersed ORC/MCM profile described in the manuscript. The homogenous ORC/MCM distribution over the genome was addressed by our binning approach at 1 kb resolution.

Dellino et al. performed Orc1 ChIP from low-density chromatin of HeLa cells (Dellino et al., 2013). Thereby, they introduced a bias towards early replicating and transcriptionally active euchromatic regions. Indeed, when binning Orc1 data at 1kb, Orc1 was also enriched at TSS (Author response image 1). Interestingly, low density input was also mildly enriched at TSS.

Orc2 ChIPs were performed by Miotto et al. in asynchronously cycling K562 cells using unfractionated chromatin and found to majorly depend on chromatin accessibility (Miotto et al., 2016). They detected Orc2 in promoters and regions enriched for active chromatin marks, hence also reporting a link between Orc2 and transcription. This was confirmed by binning their data at 1kb and analyzing their distribution across genes (Author response image 1).

Both Orc1 and Orc2 were found at TSS, in line with our data (Figure 3A). However, both approaches did not detect the depletion from the gene body. This observation may either result from G1 cell cycle enrichment, or technical differences, or a combination of both. TSS also seem to be robust “storage sites” of ORC, independent of the cell cycle. We also assume that rigorous sonication reduces “weak” and disperse binding throughout the genome, thus mainly detecting majorly robust ORC binding at accessible sites. Technical differences, as well as the different cell lines and cell cycle stages may be the reason why we observe a poor correlation between Orc1 (Dellino), Orc2 (Miotto) and our ORC data at 1 kb resolution (Author response image 1).

However, when we performed MACS2 peak calling on Orc2 in K562 (using default settings and cutoffs results in 16,767 detected peaks overlapping from two replicates) and calculated the average profile of our Orc2 or Orc3 at those peaks, we observe substantial co-enrichment (Author response image 1).

We included Author response image 1 in the manuscript as Figure 1—figure supplement 3e and discussed our Orc2/3 enrichments at K562 Orc2 peaks together with the relation to DNase HS sites.

**Author response image 1. sa2fig1:** Comparison of our ORC/MCM ChIPs with previous Orc1 and Orc2 ChIP-seq data. a) HeLa Orc1 and b) K562 Orc2 normalized read frequencies around TSSs or TTSs (independent of gene activity). Only genes larger than 30 kb without any adjacent gene within 15 kb were considered. Distances from TSSs or TTSs are indicated in kb. Means of Orc1 and Orc2 frequencies are shown ± 2 x SEM (lighter shadows). The dashed grey horizontal line indicates relative read frequency 1.0 for reference. c) Heat map of Pearson correlation coefficients R of our ORC ChIPs and the sum of two Orc2 replicates in K562 (Miotto et al.) and Orc1 in HeLa (Dellino et al., 2013) at 1 kb resolution. Column and line order were determined by complete linkage hierarchical clustering using the correlation distance (d = 1-r). d) Average ORC relative read frequencies at Orc2 (Miotto et al., 2016) peaks (>1 kb).

2) Figure 5A and Figure 4—figure supplement 2B. The results show that ORC is 1.4 times more frequently found in early versus late replicating regions. It is possible that the chromatin in early replicating regions is more accessible to the ChIP procedure than late replicating regions, which are likely more compact and hence difficult to access using antibodies. How have the authors excluded the possibility that extraction of DNA fragments in early versus late replicating regions could explain the difference in ORC binding? It should be noted that the authors previous papers and the Discussion in this paper claims accessibility to chromatin by replication factors may explain replication timing, yet they have assumed that the sonication and antibodies used for ChIP analysis are equally accessible. It is known that heterochromatic regions of the genome form phase transitions that may behave completely differently than actively transcribed and "accessible" regions of the genome.

The reviewers are addressing a very important issue that is often neglected when discussing ChIP results and is also often leading to misunderstanding. Late replicating regions of the genome are mainly heterochromatic and therefore often described as more compact and less accessible. This describes the situation within a cell. As a consequence, this chromatin is often more difficult to solubilize by sonification and/or MNase digest and therefore underrepresented in ChIP samples. In the ChIP test tube, however, the solubilized heterochromatin is equally accessible to antibodies as euchromatin. Furthermore, in our experiments, early replicating, “accessible” (eu)chromatin is in fact underrepresented in the input compared to late replicating (hetero)chromatin (Figure 4—figure supplement 2B). This is a consequence of the ultrasonication step: Less compact euchromatin requires less power to be fragmented to smaller fragments than compact heterochromatin. These euchromatic fragments are often too small to be captured resulting in an underrepresentation of euchromatin in ChIP samples. Therefore, input normalization enhances, rather than negates, the observed ORC enrichment in early replicating regions. In conclusion, neither an easier chromatin sonication nor an easier access of solubilized chromatin to antibodies can explain the stronger binding of ORC to early replicating regions.

3) Figure 5E. The same concern outlined in comment 2 above could explain the small, albeit statistically significant difference between H4K20me3 high and low regions of the genome.

We do not have any information about the relative accessibility of H4K20me3-high/low regions to provide a direct answer to this point. However, both H4K20me3-low and -high data sets are extracted from late replicating domains, as we are exclusively investigating late-replicating, non-genic AS and flat, null RFD (NRRs) that are only found in late-replicating DNA. Therefore, we do not expect a differential cellular accessibility and solubility unless uncorrrelated to replication timing.

Furthermore, analyses of ORC/MCM enrichments at H4K20me3-low and H4K20me3-high sites in G1 and S-G2-M chromatin clearly show the dynamics of ORC/MCM before and after replication. In particular ORC/MCM binding is reduced in S-G2-M at the high-H4K20me3 windows, confirming the expected dynamic behavior at potential replication origins.

We have integrated this argument in our manuscript and the S-G2-M data set as Fiugre 5F and Figure 5—figure supplement 1F.

4) Figure 6A and related text. The very slight differences in H4K20me3 levels could also be explained by extraction artIfact.

We are working with up to three biological replicates and target 6 different proteins (Orc2, Orc3, Mcm3, Mcm7, H4K20me3, H4K20me1). Extraction artefacts specific to early versus late replicating chromatin would appear in all samples, disregarding the targeted protein. In the case of H4K20me3, we observe a specific enrichment of high H4K20me3 levels in late replicating NRRs (Figure 6A, bottom row) that is not observed for any ORC/MCM, hence we can almost certainly exclude that our observation relies on extraction artefacts.

5) “However, potential origins are defined by assembled MCM-DHs, not by ORC”. This statement that potential origins are determined by the MCM2-7 DH and not by ORC is not logical because MCM2-7 DH is loaded by ORC and other factors. The idea that it correlates with ORC is dismissed a few lines later, but none of these statements are justified.

We agree that MCM-DH is loaded by ORC. However, it is the MCM-DH helicase that eventually nucleates replisome assembly following duplex melting (Yeeles et al., 2015). Therefore, the sites of replication initiation are ultimately dictated by the location of activated MCM-DH, not ORC. Accordingly, it has been shown several times that ORC is dispensable for replication initiation once MCM-DH have been loaded (Gros et al., 2015; Hua and Newport, 1998; Rowles et al., 1999). Furthermore, once loaded onto DNA by ORC, the MCM-DH helicase is free to either diffuse away from ORC (Evrin et al., 2009; Remus and Diffley, 2009) or be actively displaced by transcription or other chromatin-related processes (Edwards et al., 2002; Gros et al., 2015; Powell et al., 2015; Ritzi et al., 1998) Therefore, we expect that the location of initiation sites is defined by activated MCM-DH, whose location can differ from ORC.

What is the evidence that access of firing factors to MCM2-7 DH are regulated by chromatin access?

A large number of studies correlate genome-wide DNA accessibility (as measured by nuclease, methylase or transposase assays) with early replication and/or sites of replication initiation (Bell et al., 2010; Gilbert et al., 2004). We assume that the reviewers do not question this correlation. The question is therefore whether this correlation reflects causality.

On the one hand, we see no reason to assume that limiting origin firing factors would diffuse in chromatin very differently from the proteins used in DNA accessibility assays. We therefore expect them to encounter MCM-DH in accessible chromatin faster than in less accessible chromatin, so that even a genome with a theoretical strictly uniform density of MCM-DH would still show early- and late-replicating regions. Thus, replication timing would be at least in part the inevitable consequence of the uneven accessibility of the genome.

On the other hand, it has been proposed that budding yeast replication timing profiles can simply be accounted by a "multiple initiator model" in which all MCM-DH are equally accessible to limiting firing factors but individual origins differ by the number of bound MCM-DH so that origins with more MCM-DH fire earlier on average than those with less (Yang et al., 2010). However, Das et al. have reported that the correlation between MCM ChIP-seq signal and timing, although significant, is < 0.5 and that other factors must contribute as well (Das et al., 2015). For example, large numbers of MCM-DHs are loaded at late-replicating telomeres, suggesting that heterochromatin can somehow delay the firing of loaded MCM-DHs.

Histone modifications are known to modulate chromatin accessibility with effects on origin firing time. In budding yeast, moving a given origin from a late to an early replicating region of the chromosome can advance its firing time (Ferguson and Fangman, 1992). Targeting histone acetylases or deacetylases to specific chromosomal sites can increase or decrease origin efficiency both in yeast (Vogelauer et al., 2002) and human cells (Goren et al., 2008). Deleting the yeast Sin3-Rpd3 deacetylase causes origins that normally fire late to fire earlier in S phase, coincident with increased acetylation and accelerated loading of Cdc45, a limiting origin firing factor (Aparicio et al., 2004; Vogelauer et al., 2002). Thus, the local chromatin environment has the ability to potentially permit or restrict an origin from firing.

One potentially important difference between protein samples used in accessibility assays and diffusible origin firing factors is that the latter may experience specific physical interactions that modulate the effect of chromatin accessibility. For example Swi6, a fission yeast HP1 homologue, interacts with DDK, a rate-limiting origin firing kinase, to activate origins in early S phase, specifically at pericentromeric and silent mating-type heterochromatin loci (Hayashi et al., 2009). Mutations that diminish this interaction but do not affect Swi6/HP1 localization result in retardation of replication specifically at these heterochromatin loci. Therefore, this specific interaction can overcome the suppressive effect of “closed” chromatin created by Swi6/HP1. Similarly, a prevailing hypothesis is that chromatin acetylation stimulates DNA replication by opening chromatin structure (Gindin et al., 2014). However, recent results suggest that the acetyl-histone binding proteins BRD2 and BRD4 physically interact with the rate limiting origin firing factor TICRR/TRESLIN and that abrogation of this interaction disrupts the normal replication program (Sansam et al., 2018).

In summary, it is clear that the replication program reflects both, heterogeneity in MCM-DH loading and regulation of MCM-DH activation by chromatin structure. Whether specific histone modifications act by facilitating diffusion of firing factors through chromatin, by providing specific docking platforms for such factors, or by a combination of both mechanisms is not completely elucidated. At any rate, however, heterogeneity in MCM-DH loading alone appears insufficient to explain our results. An additional, significant role for chromatin structure in regulating MCM activation after MCM loading by ORC therefore appears likely. We have restructured the corresponding paragraphs in the Discussion, cited selected relevant studies, and tried to integrate these complex mechanisms in a rephrased Conclusion and the graphical model (Figure 7).

6) In a significant paper describing ORC ChIP and replication initiation, it was shown that ORC binding correlates with histone H2AZ and this could explain early replication origin activity. This paper is not even cited, much less discussed, and it should be (see Long et al., 2019.

We apologize for omitting this important paper. We have now discussed this publication in our manuscript. Its findings are in line with our previous paper that showed H2A.Z enrichment within AS (Petryk et al., 2016).

7) The authors have dismissed the replication timing model proposed by Miottto et al. 2016, but it is not clear why. This model should be discussed in relationship to the model in Figure 7.

We had no intention of dismissing the replication timing model proposed by Miotto et al. (Miotto et al., 2016). The authors simulated in this study replication timing based on their detected ~52,000 Orc2 peaks and assuming stochastic origin firing. They obtained a good correlation between simulated and Repli-seq determined replication timing profiles, and the correlation improved when they also assumed dispersed origin firing from “non-specific” Orc2 sites, that escaped experimental detection in their setup. Consistent with Miotto et al., we observed global differences of ORC levels when comparing early against late RTDs (Figure 5A, Figure 4—figure supplement 2B). In addition, we detected ORC in regions where Miotto et al. suggested its presence but did not detect it.

However, in an independent modelling approach (Gindin et al., 2014), all chromatin marks associated to open chromatin also allowed very good predictions of replication timing based on a large diversity of chromatin marks associated to open chromatin. Hence, the question remains of what is causally responsible for early replication / replication initiation. Furthermore, the current resolution of RT profiles (50-100 kb) is not sufficient to map replication IZs, whereas the resolution of RFD profiles (1-5 kb) does. Whether the models of Miotto et al. or Gindin et al. remain valid up to replication initiation sites also remains to be addressed.

Our work, which for the first time compares the location of ORC and MCM to high-resolution RFD profiles, suggests that neither ORC nor MCM density suffices to predict replication initiation at this resolution. We therefore conclude that the replication program reflects not only heterogeneity in ORC/MCM-DH density but also regulation of MCM-DH activation by chromatin structure (see additional comment 5 above). In other words, our model is consistent with the model of Miotto *et al.,* but with an additional layer of regulation. We added a paragraph in the Discussion to explicit these points.

[Editors' note: further revisions were suggested prior to acceptance, as described below.]

The revised paper has incorporated new data that compares the abundance of ORC2, ORC3, MCM3 and MCM7 protein at two difference stages of the cell division cycle and raises some interesting observations. The authors have extensively addressed all of the original reviewer comments and provide new analysis. The differences in MCM and ORC levels at the different classes of gene expression, except for the gene bodies, are very modest but nonetheless statistically significant.The general conclusion is that ORC is wide spread on the genome in G1 phase and MCM localizes with sites of initiation of DNA replication, and that histone H3K4me3 is correlated with late origin firing. At all locations, the exact site of initiation of DNA replication is stochastic.One paradox needs explaining that arises from the new data presented in the revised paper compared to data in the literature.1) The data in Figure 4—figure supplement 2C show that MCM3 and MCM7 levels are reduced in S-G2-M cells compared to G1 cells, but there remains a difference between early versus late replication timing domains in both cell cycle stages. In contrast, ORC is high in early RFDs and low in late RFDs at both stages. Perhaps the authors should discuss the significance of this result, in light of the fact that ORC1 is degraded in human cells at the G1-S transition and should not be present later in the cell cycle util it is re-synthesized. Does this mean ORC2 and ORC3 remain chromatin bound during the cell cycle and what does this mean.We suggest that the authors address this issue in the Discussion of a revised manuscript which can then proceed.

We thank the reviewer for raising this interesting question. It is well described that Orc1 in human cells is degraded at the G1-S transition and in early S phase and is re-synthesized at in mitosis. However, the chromatin binding of the remaining ORC subunits is controversially discussed. In our study, we detect Orc2 and Orc3 binding to S-G2-M chromatin. However, ChIP-seq, which we used to detect chromatin-bound ORC only allows monitoring the relative distribution of chromatin-bound proteins along the genome and not their absolute levels. Therefore, we can only be speculative and do not exclude that Orc2 and Orc3 binding to chromatin is globally decreased after replication. The binding of Orc2 and Orc3 we detect in S-G2-M may either occur independently of Orc1 or reflect the binding of the entire complex in late mitotic cells. We have added a corresponding discussion in our manuscript.